# Organo–organic and organo–mineral interfaces in soil at the nanometer scale

Angela R. Possinger [1,6], Michael J. Zachman [2,7], Akio Enders[1,8], Barnaby D. A. Levin [2],
David A. Muller [2,3], Lena F. Kourkoutis [2,3] & Johannes Lehmann [1,4,5 ✉]

The capacity of soil as a carbon (C) sink is mediated by interactions between organic matter and mineral phases. However, previously proposed layered accumulation of organic matter within aggregate organo–mineral microstructures has not yet been confirmed by direct visualization at the necessary nanometer-scale spatial resolution. Here, we identify disordered micrometer-size organic phases rather than previously reported ordered gradients in C functional groups. Using cryo-electron microscopy with electron energy loss spectroscopy (EELS), we show organo–organic interfaces in contrast to exclusively organo–mineral interfaces. Single-digit nanometer-size layers of C forms were detected at the organo–organic interface, showing alkyl C and nitrogen (N) enrichment (by 4 and 7%, respectively). At the organo–mineral interface, 88% (72–92%) and 33% (16–53%) enrichment of N and oxidized C, respectively, indicate different stabilization processes than at organo–organic interfaces. However, N enrichment at both interface types points towards the importance of N-rich residues for greater C sequestration.

[1] Soil and Crop Sciences, School of Integrative Plant Science, Cornell University, Ithaca, NY 14853, USA. [2] School of Applied and Engineering Physics, Cornell University, Ithaca, NY 14853, USA. [3] Kavli Institute at Cornell for Nanoscale Science, Cornell University, Ithaca, NY 14853, USA. [4] Institute for Advanced Study, Technical University of Munich, Garching, Germany. [5] Cornell Atkinson Center for Sustainability, Ithaca, NY 14853, USA. [6] Present address: Department of Forest Resources and Environmental Conservation, Virginia Tech, Cheatham Hall, Blacksburg, VA 24060, USA. [7] Present address: Center for Nanophase Materials Sciences, Oak Ridge National Laboratory, Oak Ridge, TN 37831, USA. [8] Present address: Department of Biological and Environmental Engineering, Cornell University, Ithaca, NY 14853, USA. ✉email: CL273@cornell.edu

Soil organic carbon (SOC) constitutes a critical reservoir in the global C cycle, which highlights the importance of understanding the processes that drive soil organic matter (SOM) persistence, ranging from global (e.g., climate) to very fine scales (e.g., organo–mineral surface interactions)[1–4]. Improving the ability to describe drivers of SOM persistence, including mechanisms of SOM protection, enables better prediction of changes in the soil C reservoir in light of global environmental change[5].

Soil OM is a mixture of organic compounds subject to organo–mineral interactions[6–10]. Such interactions between SOM and mineral phases result in lower microbial accessibility and availability for decomposition, which is seen as a dominant process for SOM stabilization[4,9,10]. The spatial and chemical heterogeneity of SOM, soil physical structures, and microbial distribution at the scale of soil micro- and macroaggregates and pore structures are relatively well studied[7,11–16] (Fig. 1a). In contrast to microaggregate-scale heterogeneity, previous imaging and spectroscopy of SOM coatings within micrometer-scale organo–mineral assemblages have shown relatively uniform, ordered layers with distinct OM composition as a function of distance from mineral surfaces at smaller spatial scales of micrometers[15–19] (Fig. 1b). The spatial architecture of layered SOM accumulation is consistent with the concept of preferential association of nitrogen (N)-bearing and oxidized functional groups at the mineral surface, acting as a backbone for additional layers of OM accumulation with different composition (i.e., the zonal-structure model)[20,21]. While mono-layer adsorption of SOM on mineral surfaces and the maximum OM accumulation (i.e., saturation) could be limited by mineral surface area and chemistry, layering of OM may circumvent these limitations and therefore be effective for maximizing soil organic C sequestration[13,22].

However, the suite of interactions among OM constituents (e.g., biomolecules, microbial cell membranes, extracellular products, and small soluble compounds)[23] in addition to mineral associations challenges previous observations of ordered layers of OM accumulation. The resolution of previously used imaging and spectroscopy techniques (e.g., 30–50 nm in ref. [17]) may be too coarse to resolve or describe the interfaces among OM constituents embedded within an organo–mineral assemblage (Fig. 1b). Despite the potentially critical role of interactions between OM forms in shifting the current view of the spatial arrangement of SOM accumulation on mineral surfaces, the chemical composition of these interfaces has not been directly visualized or described in natural soil samples at the relevant nanometer scale.

Associations between SOM and semicrystalline reactive iron (Fe) and aluminum (Al) mineral surfaces are recognized to contribute to long-term SOM persistence and accumulation across widely variable soil types[8,24,25]. The formation of associations between reactive Fe and Al and SOM has been linked to preferential reactions with oxidized functional groups[26] and N-containing biomolecules[21,27–29]. However, given the sub-micrometer spatial scale of OM distribution and its chemical complexity[7], little spatially explicit evidence exists to understand these hypothesized mechanisms of interactions in natural soils at the scale of single (<10) nanometers. Further, analysis of the spatial architecture at the necessary resolution to detect the potential effect of interactions among OM phases with different compositions is lacking. Single-digit nanometer-scale imaging and spectroscopy techniques may enable us to confirm the existence of previously hypothesized nanoscale interactions, and generate novel hypotheses to be tested at larger scales based on observations of previously inaccessible structures.

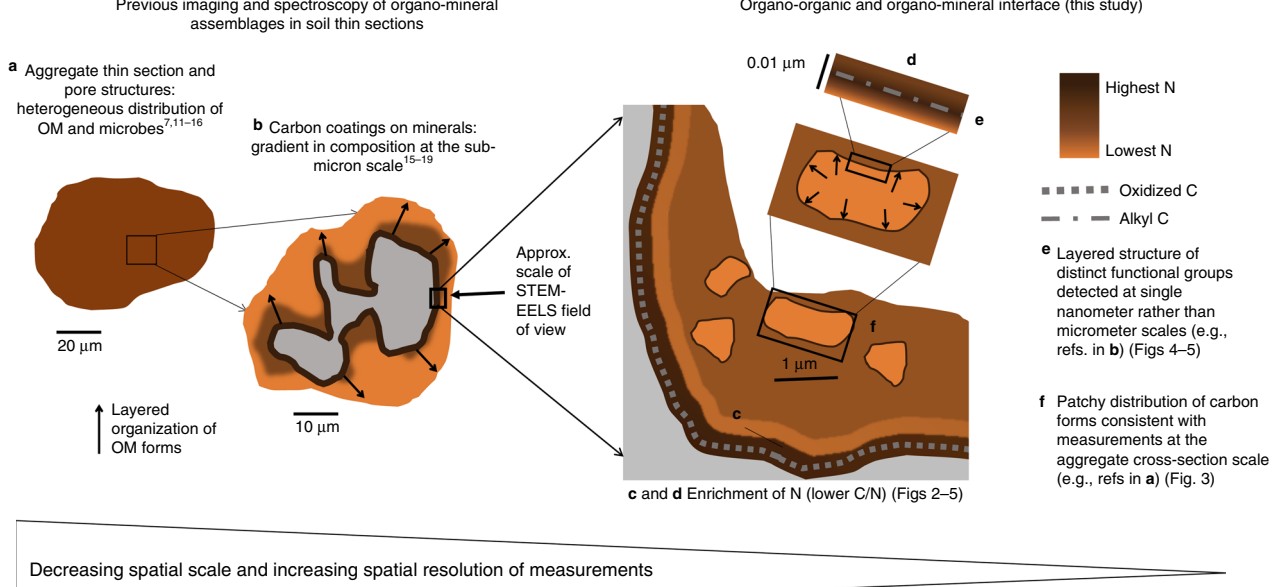

**Fig. 1 Conceptual summary of organo–organic and organo–mineral interactions.** Shifting conceptual view of the spatial architecture and composition of organo–mineral and organo–organic interactions. **a** At the scale of aggregate thin-sections, heterogeneous distribution of organic matter (OM) and microbial populations is well-known. **b** However, previous imaging and spectroscopic characterization of organo–mineral assemblages using a 10–50 µm field of view has shown organized OM layers of different composition. The concept of layered OM accumulation (i.e., the zonal model) has often been linked to nitrogen (N) enrichment at the mineral surface[20,21]. **c–f** The images and spectroscopic data presented in this study (conceptualized in **c–f**) show an order of magnitude smaller spatial scale. **c** Enrichment of N was detected at an organo–mineral interface in a volcanic soil sample (Fig. 2). **d–f** Irregular organic structures and the organo–organic interface between them were also detected (in a separate volcanic soil sample) (Figs. 3–5). These observations directly reveal heterogeneous patches that contradict previous assumptions of ordered and unidirectional layering of OM forms, generating new insights into OM composition at mineral–organic and organo–organic interfaces. Note: scale bars are for illustrative purposes only.

The goal of this study was to examine the hypothesized zonal-structure model of interaction and identify the functional groups at the organo–mineral interface. To gain further insight into the properties of organo–mineral interactions in natural soil samples and to ascertain the presence of what we call organo–organic interactions, we developed a novel approach for cryogenic sample preparation in conjunction with analytical cryogenic scanning transmission electron microscopy and electron energy-loss spectroscopy (cryo-STEM-EELS). This approach pairs single-digit nanometer spatial resolution with the ability to resolve variations in the C bonding environment across organo–mineral and organo–organic interfaces. This approach also avoids the use of C-based stabilizing resins that typically make interpretation of the native soil C content and bonding environments difficult, and enables direct visualization and analysis of the interfaces between organic and mineral phases in soil without alteration of the spatial architecture. We applied cryo-STEM-EELS to probe organo–organic and organo–mineral interfaces in archetypical high-reactive Fe and Al volcanic soils (Supplementary Table 1) with a known capacity for long-term C accumulation[8,26,30].

## Results

**Organo–mineral interface: enrichment of N and oxidized C.** Our observations of an organo–mineral interface at the nanometer scale provide direct evidence for the role of N-containing functional groups of polar C compounds in interactions at aluminum (Al) mineral surfaces. Using cryo-STEM-EELS, we show enrichment of C K-edge EELS features at 288.1 eV, indicative of oxidized carboxylic/carbonyl C with substituted or adjacent N[31,32] (Fig. 2a, b). The average intensity of oxidized C (286.6–289.0 eV) relative to lower-energy C (284.0–286.5 eV) intensity increased by approximately 33% (range: 16–53%) at the organo–mineral interface compared to the adjacent OM (Fig. 2, Supplementary Table 2). Relative enrichment of N was indicated by a decrease in the C/N signal intensity (approximately 88% decrease, range: 72–92%) closer to the Al mineral surface compared to an adjacent OM region (Fig. 2, Supplementary Table 2; Al content and structures in Supplementary Figs. 1 and 2).

The presented data provide unambiguous evidence of co-location between an Al mineral and N-rich oxidized organic matter but do not directly probe the mechanism of interaction. However, the presence of N-rich oxidized OM is consistent with

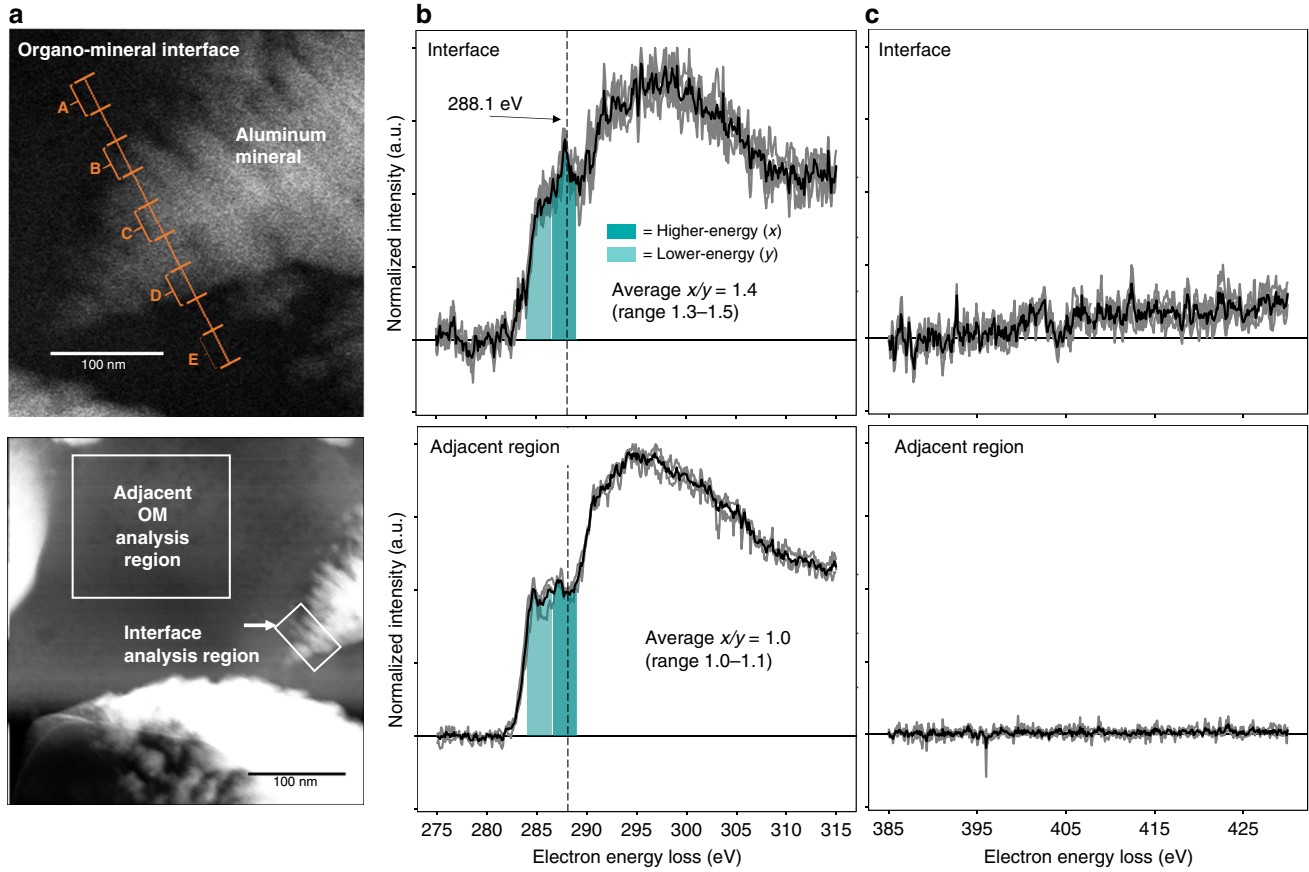

**Fig. 2 Nitrogen and oxidized carbon enrichment at the organo–mineral interface.** High-resolution cryo-electron imaging and spectroscopy of an organo–mineral interface in a volcanic soil sample. **a** Annular dark-field (ADF) scanning transmission electron microscopy (STEM) image of organo–mineral interface and adjacent organic matter region, showing the location of electron energy-loss spectroscopy (EELS) data collection across and adjacent to an aluminum (Al) mineral with layered structure. **b**, **c** Average carbon (**b**) and nitrogen (**c**) K-edge EELS (black line) and individual spectra (gray lines) for either five points (A–E) across the organo–mineral interface or three EELS point scans within the adjacent OM region. For both **b** and **c**, spectra are shown unsmoothed and normalized to the maximum carbon intensity. In **b**, dotted vertical line indicates peak at 288.1 eV, putatively identified as carboxylic/carbonyl groups with substituted or adjacent N[31,32]. Lower (284.0–286.5 eV) (*y*) and higher (286.6–289.0 eV) (*x*) energy regions are indicated by shaded boxes. Ratio *x*/*y* indicates the ratio of integrated EELS intensity within each region, normalized to total carbon integrated area (280.0–315.0 eV). Between the adjacent region and interface, the *x*/*y* ratio increased by an average 33% (Supplementary Table 2), indicating an increase in oxidized C at the organo–mineral interface compared to the adjacent C region. In addition, the ratio of integrated total carbon (280.0–315.0 eV) to total nitrogen (395.0–430.0 eV) decreased by an average 88% between the adjacent C region and the organo–mineral interface region.

the preferential retention of N-containing functional groups and oxidized (e.g., carboxylic acid) OM in soils with short-range ordered (SRO) Fe and Al mineralogy. Aromatic and carboxylic acids have been identified as key C forms in stabilizing interactions with reactive metals not only in archetypical soils with SRO mineralogy (e.g., the studied Andisols[26]), but also in broadly distributed forest soils[33]. While our data do not address the frequency or distribution of organo–reactive metal interactions, reactive metal phases are present in most soil types as surface coatings on silicate clays and primary minerals, Fe precipitates, and Fe–OM co-precipitates under fluctuating redox conditions[10]. In addition, spatial associations of N-rich OM on reactive metal surfaces have been identified both within SRO-rich soils[34] and temperate soil organic horizons[28]. More broadly, the role of N in organo–mineral associations is consistent with the role of microbially derived residues for the accumulation of OM, a foundation of the microbial efficiency-matrix stabilization (MEMS) framework[35] and mineral–microbe biogeochemical models[5]. Multiple mechanisms of N-rich OM interaction with reactive Fe and Al minerals have been proposed, including indirect association via reactive phosphate groups in phosphorylated proteins and hydrogen bond formation[20,28].

While inner-sphere ligand exchange is well documented for oxidized carboxylic and aromatic acids[26], ligand exchange

mechanisms specific to N-substituted carboxylic groups are less well-known. In this study, the very high enrichment of total N (88% lower C/N intensity ratio) at the interface also suggests that the N-rich OM association may co-occur with the accumulation of inorganic N (e.g., electrostatic retention of $NH_4^+$). The potential for either competitive or synergistic interactions between organic and inorganic N at reactive metal surfaces provides a new direction for the evaluation of N in OM stabilization. In addition, interactions between N-rich OM and reactive Al (as identified here) have received less attention than N-rich OM and reactive Fe[28,29]. With an emerging focus on Al-mediated OM stabilization under conditions of Fe depletion[34,36], further experiments should target possible divergence in physicochemical mineral–organic mechanisms between N-rich OM and reactive Al in contrast to Fe.

**Hierarchy of spatial heterogeneity in OM composition.** Cryo-STEM-EELS analysis of an OM-rich region in a soil aggregate with high OM content (Supplementary Table 1) revealed 0.1–1 µm-size features of distinct OM composition using nanometer-scale spatial resolution (Fig. 3a–c, Supplementary Fig. 3). The presence of irregular aromatic-rich C, lower-N, and higher-O organic features with distinct C composition embedded in a more

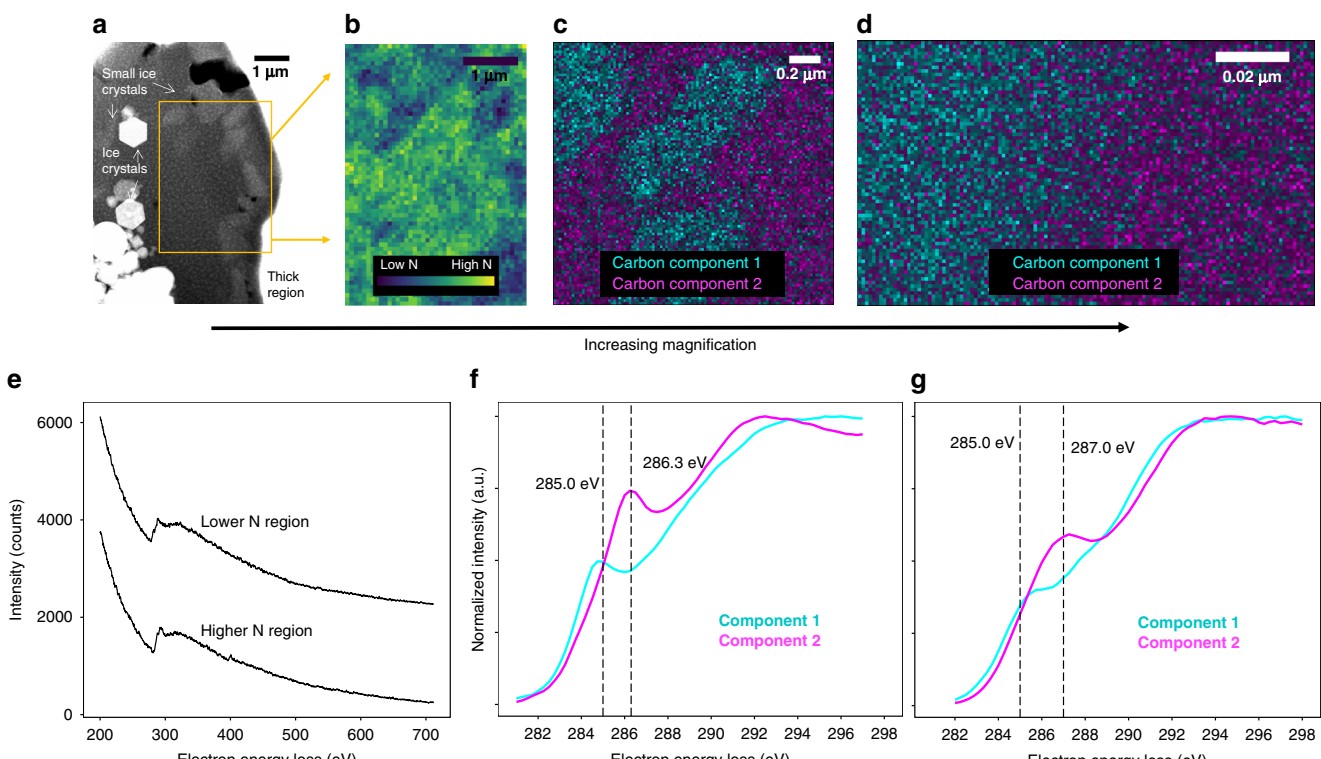

**Fig. 3 Composition of organic features and the organo–organic interface.** High-resolution imaging and spectroscopy of organo–organic features and their interface by cryo-scanning transmission electron microscopy and electron energy-loss spectroscopy (cryo-STEM-EELS). **a** Overview low-magnification (5 kx) annular dark-field (ADF) STEM image of cryo-focused ion beam (cryo-FIB) thin section of organic-rich volcanic soil. The approximate region indicated by a box was used for subsequent electron energy-loss spectroscopy (EELS) mapping of carbon (C) and nitrogen (N). **b** EELS elemental map showing the patchy distribution of low-N features in the high-N matrix. **c** Patchy distribution of organic matter (OM) features micrometers in size, shown with EELS maps of the two distinct carbon (C) bonding environments, as revealed by carbon K-edge multivariate curve resolution (MCR) component fits (component spectra are shown in **f**). **d** Nanometer-scale EELS C K-edge MCR component map of the interface between two OM patches (component spectra shown in **g**). For **c**, **d**, the cyan color channel contrast and brightness were auto-adjusted in ImageJ[71]. **e** Raw EELS data for representative lower N (top spectrum) and higher N (bottom spectrum) regions in (**b**). Spectra show the high C K-edge (~280.0 eV) intensity relative to N (~400.0 eV) and O (~530.0 eV) intensities. **f**, **g** Normalized (maximum = 1) C K-edge MCR component spectra corresponding to (**c**, **d**), with a distinctive transition from ~285.0 eV in component 1 (assigned to aromatic C=C bonds) to higher-energy features (~286.3–287.0 eV) in component 2. Features in spectra are shown by dotted vertical lines. Spectra are shown as raw (unsmoothed) MCR fit outputs normalized to the maximum output value.

alkyl-rich C with higher-N and lower-O organic matrix contradicts previous observations of micrometer-sized ordered gradients as a function of distance to mineral surfaces[15–19] (Fig. 1b).

EELS C K-edge fine structure varied greatly between the features and the surrounding matrix, which were identified as discrete C forms using multivariate curve resolution (MCR) (Fig. 3c, f). Within the observed patchy features, the EELS fine structure is dominated by a low-energy feature at ~285 eV, which is assigned to the C 1s–$\pi^*_{C=C}$ transition of C=C bonds associated with aromatic structures[32]. In contrast, the EELS fine structure of the surrounding matrix distinctly shifts from ~285 eV to higher

energy (Fig. 3f, g). At higher spatial resolution (approximately 2 nm) (Fig. 3d), the higher-energy feature is at ~287 eV (Fig. 3g, Fig. 4d). While many overlapping transitions occur in this region, features can be broadly linked to alkyl C-H bonds with a variety of transitions (e.g., C 1s–$\pi^*_{C-H}$)[32,37].

By observing C/N, C/O, and aromatic/alkyl C ratios across the boundaries of the OM patches, the improved spatial resolution of cryo-STEM-EELS over previously utilized techniques (such as scanning transmission X-ray spectroscopy with near-edge X-ray absorption fine structure[15–19]) enabled the discovery of nanometer-scale ordered gradients of layered OM at the boundary

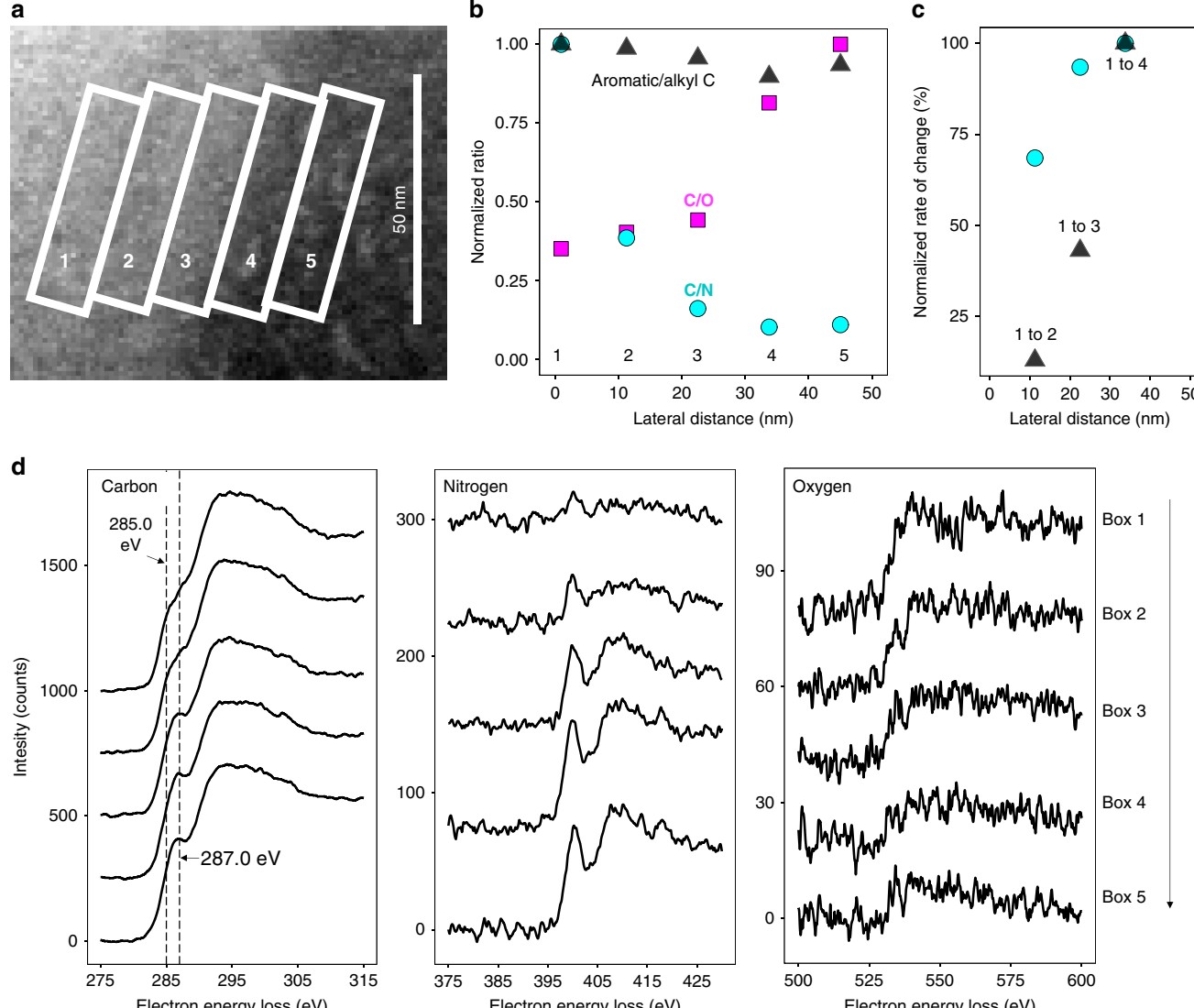

**Fig. 4 Gradient in organic composition across the organo–organic interface.** Distinct combinations of carbon/nitrogen (C/N), carbon/oxygen (C/O), and aromatic/alkyl C ratio across the organo–organic interface shows a gradient of organic matter (OM) composition at the ~50 nm scale. **a** Annular dark-field (ADF) scanning transmission electron microscopy (STEM) detail of the interface between organic phases in a soil thin section. Boxes 1 through 5 indicate regions used to compute average electron energy-loss (EEL) spectra across the interface (lateral distance). **b** Ratios of normalized (maximum = 1 unit) elemental integrated area for C/N, C/O, and aromatic/alkyl C across the interface. The aromatic/alkyl C ratio is the ratio of integrated intensity in EELS spectral regions (defined here as aromatic from 284.25 to 285.75 eV and alkyl from 286.0 to 287.5 eV) normalized to total C integrated area (280.0–315.0 eV). Each location (1–5) across the interface shows a different combination of C/N, C/O, and aromatic/alkyl C. At point 4, alkyl C is enriched relative to the high-alkyl C matrix. The calculations for each point are from ~540 individual spectra (1 spectrum per pixel) in each 11 × 44 nm box. **c** Rate of change for C/N and aromatic/alkyl ratios across the organo–organic interface relative to box 1. The C/N ratio decreases more rapidly than the aromatic/alkyl ratio for the transitions from boxes 1 to 2 and 1 to 3, which suggests trends in C composition and N quantity are decoupled across the interface. **d** Average EEL spectra (spline curve, an average of ~540 individual spectra) for boxes 1–5 for the carbon, nitrogen, and oxygen K-edges. Carbon K-edge EELS show similar intensity but a change in fine structure. For carbon, vertical lines at 285.0 and 287.0 eV are putatively associated with aromatic and alkyl C, respectively. Nitrogen and oxygen K-edge EELS show an increase and decrease in intensity across the interface, respectively.

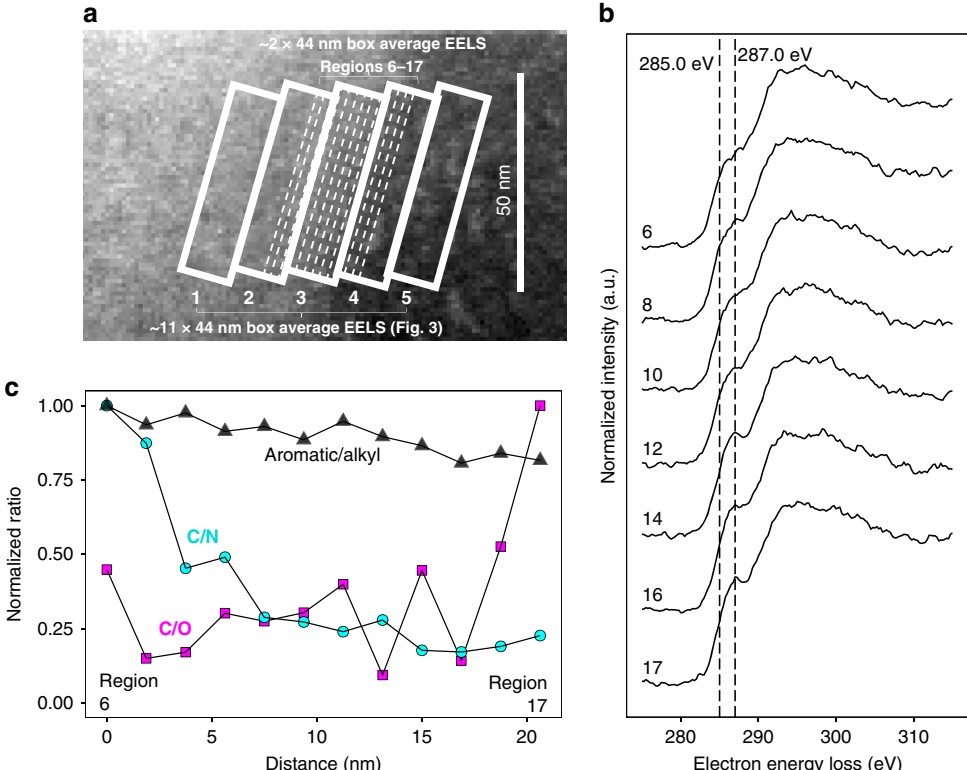

**Fig. 5 High-resolution characterization of the organo–organic interface.** Organo–organic interface characterization at 2-nm lateral resolution shows variations in organic matter composition at 5–11 nm distances. **a** Overview annular dark-field (ADF) STEM image showing approximate locations of 2 × 44 nm analysis regions (white dashed lines, labeled 6 through 17) across the interface, within 11 × 44 nm box analysis regions (1 through 5) (also shown in Fig. 4). **b** Average (2 × 44 nm boxes, ~90 individual spectra per box) electron energy-loss (EEL) spectra (normalized to maximum = 1), with vertical dashed lines at 285.0 and 287.0 eV, putatively associated with aromatic and alkyl C, respectively. **c** Aromatic/alkyl C, C/N, and C/O signal intensity ratios across the interface show fluctuations at <5 nm distances for aromatic/alkyl C ratios and <11 nm distances for C/N and C/O ratios, consistent with a layered gradient in OM composition.

of micrometer-sized organic features, in addition to micrometer-scale patchiness (Figs. 4 and 5). Along the edge of the interface, the change in aromatic/alkyl C ratios constrains the homogeneity of the interface to ~25 nm in the vertical direction (Supplementary Fig. 4).

To discern the presence of ordered gradients at this fine scale, we examined changes in C forms using ~2 × 44 nm analysis regions across the interface (Fig. 5). The transition between aromatic and alkyl C was gradual, and overall, tracked the 11 × 44 nm analysis region EELS data (Supplementary Fig. 5). The presence of clear fluctuations in aromatic/alkyl, C/N, and C/O signal intensity between adjacent 2 × 44 nm regions also suggests that ordered gradients of C composition may occur even at a scale of <5 nm, but improved spectrometer detection (e.g., direct electron detection) may be needed to definitively identify the points of separation between layers. The observed layers of distinct OM composition are consistent in size with small biomolecules or associations[23,38] of small biomolecules, such as many proteins. However, even the single-digit nanometer resolution does not allow for subnanometer spatial mapping of individual functional groups or bonds.

**Organo–organic interface: enrichment of N and alkyl C.** We identified hitherto undescribed organo–organic interfaces as part of OM composition at a very fine scale. Within aromatic-rich patches, N contents increased towards the interface with the alkyl phase (Fig. 4). The shift from low to high N contents was spatially separate from the shift from aromatic to alkyl C, with N contents beginning to increase further from the interface (Fig. 4c). In

addition, the C/N ratio was 7% lower directly at the interface (Box 4) compared to the adjacent higher-N phase (Box 5 in Fig. 4a, b, Supplementary Table 3). This suggests N-containing components of the aromatic C phase may be preferentially involved in an interaction. Aromatic C=C groups are only thought to be responsible for interactions between organic and mineral phases in specific situations[20]. In contrast, N groups in OM are expected to associate readily with other organic constituents via electrostatic interactions or covalent bonds, due to the potential for their positive charge and electron-withdrawing character, respectively. In contrast to the mineral interface analyzed in this study, enrichment at the edge of the aromatic organic phase occurred without the presence of oxidized OM (e.g., in carboxylic C of amino acids), suggesting that N-containing aromatic compounds with a relatively high C/O ratio and alkyl constituents (e.g., cytokinins) may facilitate interactions between organic phases. However, additional characterization of the N bonding environment is needed to further refine the physicochemical mechanism of interaction.

The ratio of aromatic C (integrated 284.25–285.75 eV signal) or alkyl C (integrated 286.0–287.5 eV signal) to the total integrated C edge intensity showed enrichment in alkyl C (by 4%) occurring directly at the organo–organic interface (Box 4 vs. Box 5 in Fig. 4, Supplementary Table 3). No unique third C component at the interface was present, indicating a shifting aromatic/alkyl C ratio rather than the appearance of a new C feature (Supplementary Fig. 6). The overall trends in C/N and C/O across the interface show mixing between OM forms across a boundary of ~50 nm or less across (Fig. 4). For this analysis, we achieved a single-digit

nanometer spatial resolution (see "Methods"), suggesting that the size of the overall boundary represents a property of the interface itself rather than the instrument resolution. However, the dimension of the boundary may be overestimated due to the unknown orientation of the boundary within the <200 nm-thick thin section. The transition in C EELS fine structure (regions of interest (ROI) 1–4 in Fig. 4a) from more aromatic (~285 eV) to alkyl C (~287 eV) was sharper than the gradual change in N and O composition (Fig. 4c).

## Discussion

Taken together, these observations provide evidence for a shifting view (Fig. 1) of the hierarchy of OM spatial distribution in soil and the possible role of OM accumulation facilitated by organo–organic interactions. At the scale of a microaggregate and soil pore, OM is inherently heterogeneous in spatial distribution and composition[7,11–16]. The spatial separation of OM composition within an OM-rich region observed here underscores the need to consider nonlinear organization of organo–mineral interaction models due to patchy, disordered OM accumulation at a very fine scale. At such high resolution, we show evidence for <5–11 nm-thick layers of OM forms (Figs. 4 and 5), approaching a meaningful scale to consider the implications for the zonal model[20], though still larger than the size of individual functional groups or bonds. At this scale, our observations do not contradict the existence of discrete zones (i.e., layers of OM composition). However, we highlight that the arrangement in space may not only extend in ordered layers away from a mineral surface but also at the boundaries of irregularly shaped organic patches (Fig. 1d–f). In addition, enrichment of N co-occurred with oxidized C at the mineral interface, in contrast to alkyl C at the organo–organic boundary. This distinction suggests the need to consider a combination of convergent and unique interaction pathways of organo–organic interactions in addition to the current focus only on the variations among organo–mineral interaction mechanisms.

The existence of organic patches and organo–organic interfaces with different organic C composition than those at organo–mineral interfaces generates important information for sequestering soil organic C and predicting its changes. The described organo–organic interactions may occur at a distance to or separately from mineral surfaces, pointing toward the need to consider soil organic C stabilization mechanisms that are independent of soil minerals (de-emphasizing variables such as mineral surface area or surface chemistry), and involve N-rich surfaces[13]. In litter decomposition, organo–organic interactions between aromatic and N-rich compounds have been linked to slower litter turnover[39]. Here, we suggest interactions between relatively low and high-N OM forms may also be relevant for soil OM.

The demonstrated existence of organo–organic in addition to organo–mineral interactions may also provide an avenue to help explain that saturation of soil organic C is often not clearly predictable in the field, even though SOM saturation is a theoretically reasonable and experimentally sound expectation[40,41]. Despite the emerging evidence that N-rich OM promotes SOM accumulation[35,42], major Earth system models (e.g., the Community Land Model 5.0) treat soil N availability as primarily a limitation on decomposition[43]. Further, long-term N fertilization experiments in the field have shown mixed effects on SOM, ranging from decreased SOM decomposition rates and facilitation of occlusion under N fertilization[42] to recent work[44] showing no or inconsistent effects of N addition. Further, the distinction between SOM accumulation (i.e., amount) and persistence (i.e., turnover time) with respect to N involvement in stabilization is not well-known. The potential for system-specific (i.e., dependent on soil physicochemical properties) N effects on SOM accumulation and decomposition[44] emphasizes the need to further interrogate the biophysical mechanism of SOM persistence that may be conferred by N.

The characteristics of the identified organo–mineral and organo–organic interfaces provide an incentive to perform targeted experiments testing the biogeochemical importance of organo–organic interactions in soils, including the implications of input composition varying in N functional group composition (e.g., plant and animal manures, soil fungal and bacterial necromass or metabolites). These observations also provide motivation to expand a one-dimensional model of SOM stabilization of changing OM composition with distance to mineral surfaces to a multi-dimensional model[45,46], even at a very small scale as investigated here, recognizing that multi-phase organic matter does not align at a predictable gradient with respect to the soil mineral matrix.

However, the presented data on their own are not sufficient to quantify the persistence that N-facilitated organo–mineral or organic–organic interactions may confer to SOM. The link between reactive Fe and Al mineral abundance and increased persistence of bulk SOM is built on the assumption that mineral bonding limits microbial access to the substrate and slows OM desorption[9]. In our study, we cannot resolve the age of the OM observed (e.g., via $\Delta^{14}C$), though previous soil $\Delta^{14}C$ measurements indicate that bulk OM is very old at the studied site, typically more than 6500 radiocarbon years[47–49]. To demonstrate a difference in persistence of OM at the mineral surface requires further methodological advancement to pair high-resolution imaging with quantification of C composition and natural abundance $\Delta^{14}C$ measurements (e.g., by the advancement of monochromated EELS techniques[50,51] for damage-sensitive materials). Additionally, our results provide direct evidence only of interactions between the studied mineral and organic materials, but these techniques can in the future be applied to other soil types to evaluate their importance in other systems. Future research should also include cryo-electron tomography and the development of methods that pair ultrahigh-resolution measurements with complementary imaging at the scale of aggregate cross-sections. Further work to reveal the full three-dimensional structure and spatial context of both organo–organic and organo–mineral interfaces may ultimately inform a multi-dimensional spatial model of SOM accumulation.

## Methods

**Overview**. In this study, we identified and analyzed naturally occurring interfaces between mineral and organic phases in a natural soil sample. Our approach employed a combination of very high spatial resolution imaging by STEM and compositional analysis of organic carbon by EELS under cryogenic conditions (cryo-STEM-EELS). Due to the targeted nature of our analysis, we emphasized: (1) the likelihood of encountering such interfaces within a very small (micrometer-scale) analytical field of view (Supplementary Fig. 7), (2) how to prevent changes to the mineral–organic interface architecture due to drying or formation of crystalline ice, and (3) a method for effective preparation of the electron-transparent (<~200 nm thick) thin sections necessary for STEM-EELS.

**Study site and sample collection**. Cryo-STEM-EELS procedures to reveal organo–organic and organo–mineral interactions were conducted using subsoils with a high carbon (C) and iron (Fe) content derived from volcanic parent materials (Andisols). While Andisols occupy a relatively small land area (0.7% globally), this soil type stores a disproportionately large amount of SOC (1.3% of global SOC to a depth of 1 m)[52]. Andisols have been studied as an archetype of reactive Fe and Al control on SOM persistence for several decades. The oldest SOM in this study system is expected between 20 and 1500 kyr of soil development[8]. For this study, subsoils were collected from the Pololu Flow on Kohala, HI, with approximately 350 kyr of soil development from tholeiitic lavas, and contain high C, Fe, and Al (Supplementary Table 1, Supplementary Figs. 1–3)[34]. Subsoils were chosen for analysis due to their expected increased mineral-associated SOM content, older radiocarbon ages, the lower contribution of particulate OM, and reduced signal of land use or vegetation[2,34,53–55]. Soil pits were excavated by hand to ~1 m and sampled by the pedogenic horizon (in 2014) as described in previous

studies[26,30]. Horizon boundaries were defined by changes in color, texture, and structure as outlined in standard soil survey protocols[56] and named by top to bottom depth (e.g., 0.7–0.9 m). For each horizon, a bulk sample was collected quantitatively (i.e., from top to bottom edges to represent the average bulk composition of each horizon). Previous bulk $\Delta^{14}C$ measurements indicate a fraction modern (Fm) value of ~0.28 (in the high organic content sample) and similar values for nearby subsoils[47–49], as well as high oxalate-extractable (noncrystalline and semicrystalline) Fe and Al mineral contents (e.g., allophane, imogolite, and ferrihydrite)[34] (Supplementary Table 1). Soils were stored at field moisture conditions in coolers during sampling and transport (4–9 °C range) prior to cryo-STEM sample preparation and imaging (conducted in 2014–2017). Soil samples were not dried or frozen to prevent artifacts that may alter the architecture of the organo–mineral interface[57,58].

**Cryogenic thin-section preparation**. *Background:* The heterogeneity of soil provides a methodological challenge for preparing soil samples for high-resolution transmission electron microscopy, particularly for C, N, and other light elements easily damaged by high-energy analysis techniques. Specifically, sample preparation approaches are needed that: (1) maintain mineral–organic, organic–organic, and pore space spatial distribution, (2) eliminate the use of C-based resins that preclude the interpretation of C spectroscopic data, (3) reduce the risk of beam damage, (4) result in appropriate sample thicknesses (<100–200 nm) for electron transparency, and (5) avoid freeze–thaw and wet-dry fluctuations that alter mineral–OM interactions[57,58].

To be able to resolve native soil OM composition, thin-sectioning without the use of stabilizing materials that interfere with OM detection provides a technical challenge and avenue for methodological advancement. For nanoscale secondary ion mass spectrometry (NanoSIMS), this has been addressed by using C-based resins with distinctive isotope signatures or chemical markers[59]. However, EELS is generally not sensitive to isotope signatures at core-loss edges. Elemental sulfur (S) has been employed as a stabilizing material[60] but is sensitive to temperature changes, sublimates in the vacuum of the electron microscope at room temperature[61], and does not fully permeate the aggregate microstructure[60].

Performing sample thin-sectioning under cryogenic conditions—a common way to reduce sample damage for microscopy and spectroscopy of biological materials[62,63]—is an approach to stabilize hydrated soil samples that precludes the need for either C-based resins or elemental S[15]. Cryo-thin-sectioning followed by room-temperature imaging and spectroscopy is a common approach, though sample thawing and drying is often required[15]. To address the methodological challenges above, we developed an integrated sample preparation procedure that maintained the soil aggregate in cryogenic conditions for initial sectioning, thinning to electron transparency, and STEM-EELS mapping (Supplementary Figs. 7–9).

*Aggregate cryo-ultramicrotome pre-thinning:* Cryogenic thin-sectioning (to approximately 1–5 μm thickness) was undertaken to improve efficiency and selection of regions for subsequent cryo-focused ion beam (FIB) milling to electron transparency. Subsamples (~100 g) of bulk soils from two subsoil horizons were sieved (53–150 μm) without added pressure at field moisture to isolate natural microaggregates. This size fraction was selected because microaggregates are expected to represent the aggregate fraction with the greatest contribution to persistent carbon[64–66]. Aggregates were sparsely distributed on glass fiber filter paper (Whatman GF/A) and gradually hydrated with deionized (DI) water using an ultrasonic humidifier at its lowest setting (SPT Ultrasonic Humidifier, Sunpentown, Inc., City of Industry, CA)[15]. Single intact microaggregates were selected with a fine metal pin under a dissecting microscope and transferred to a DI water droplet on an aluminum cryo-microtome pedestal. This was inverted to encourage migration of the aggregate to the droplet surface, then rapidly cooled by immersion in slush nitrogen (N$_2$) (average −207 °C[67]) to decrease the size of ice crystals relative to a standard freezer (e.g., −20 to −80 °C). Aggregates were sectioned to a thickness of 1–5 μm with a diamond knife using a Leica EM UC7/FC7 (Leica Microsystems, Inc., Buffalo Grove, IL). The sample temperature during microtoming was kept at −60 °C. Thin sections were transferred via dry pick-up using an eyelash tool and placed on either 300 or 400-mesh, adhesive-coated, copper TEM grids, and stored in cryo-TEM grid boxes under liquid N$_2$.

*Cryo-FIB transfer:* Pre-thinned samples were transferred from storage to the cryo-FIB stage at temperatures near that of liquid N$_2$ and maintained in the cryo-FIB at −165 °C. Further details describing the cryo-transfer system are published in ref. [68].

*Cryogenic FIB Milling:* Preparation of <100–200 nm sections for STEM-EELS was completed using an FEI Strata 400 STEM DualBeam FIB (FEI Company, Hillsboro, OR) equipped with a Quorum PP3010T Cryo-FIB/SEM Preparation System (Laughton, East Sussex, UK). To maintain spatial characteristics and prevent sample loss or damage for relatively brittle and heterogeneous soil thin sections, two thinning approaches were adopted: (1) thinning with the Ga$^+$ ion beam at a shallow angle to the surface of the soil section, creating a wedge with a tapered thickness and an edge approaching zero thickness; and (2) milling both above and below the region of interest in the soil section at a shallow angle to generate a lamella of nearly uniform thickness (<200 nm) in the material (Supplementary Figs. 8, 9). Scanning EM images to monitor the milling process were collected sparingly at 5 kV, using a low beam current to minimize damage.

For milling, an ion beam voltage of 30 kV was used, with current varying between approximately 5 and 500 pA.

*Cryo-FIB sample damage assessment:* While cryogenic techniques improve sample integrity[62,63], sample damage remains a consideration for both cryogenic FIB sample preparation and subsequent cryogenic analytical electron microscopy. In order to assess improvement in sample integrity with cryo-FIB sample preparation versus using room temperature FIB, we completed the preparation and imaging of a volcanic soil sample thin-section using room-temperature lift-out FIB techniques. Initial pre-thinning to approximately 1–5 μm-thickness was completed using cryogenic ultramicrotome preparation as described for cryogenic FIB. After cryo-ultramicrotome thinning, thin sections were transferred to 400-mesh copper TEM grids with self-prepared thin C coating, brought to room temperature, and air-dried.

Samples were transferred to an FEI Strata 400 STEM FIB instrument (FEI Company, Hillsboro, OR) at room temperature. After air-drying, the aggregate cross-section broke apart along natural pore space borders, allowing for FIB lift-out of appropriately sized aggregate regions (approximately 20 × 15 × 1 μm) without on-grid cross-section (e.g., trench) milling (Supplementary Fig. 10). The aggregate region of interest was fastened to a tungsten (W) FIB lift-out needle using deposited platinum (Pt) organo-metallic alloy and transferred to a Cu TEM lift-out grid (Electron Microscopy Sciences, Hatfield, PA). After transfer, the top edge of the sample was coated with Pt for further thinning. The sample was stored dry at room temperature prior to STEM imaging in an FEI F20 S/TEM instrument (FEI Company, Hillsboro, OR) at 200 kV.

With room temperature FIB, sample redeposition during milling and distortion of sample structure was observed at both the millimeter and nanometer-scale (Supplementary Fig. 10). Higher-contrast spots are likely a result of the redeposition of sample material, or possibly implantation of gallium ions (Ga$^+$) directly from the ion beam milling. While some Ga$^+$ was detected in cryo-FIB samples (Supplementary Fig. 1), the sample distortion and higher-contrast spots observed with room-temperature FIB were not observed with cryo-FIB sample preparation (Fig. 2).

**Cryogenic EDX elemental analysis**. The elemental composition of the soil thin sections was assessed using electron dispersive X-ray (EDX) spectroscopy. For the organo–mineral interface soil sample (wedge thin section), elemental maps were collected for regions located near the primary EELS analysis region with an FEI F20 TEM-STEM instrument (FEI Company, Hillsboro, OR) equipped with an Oxford Instruments X-Max 80 mm$^2$ EDX detector (Oxford Instruments, Abingdon, Oxfordshire, UK) (Supplementary Figs. 1, 2). For the organo–organic interface soil sample (lamella thin section), an EDX point scan was collected (5 kV voltage) adjacent to the thin-section area during cryo-FIB preparation with an Oxford Instruments X-Max 80 mm$^2$ EDX detector installed on the FEI Strata FIB/SEM instrument described above (Supplementary Fig. 3).

**Cryogenic STEM-EELS**. Wedge thin-sections were transferred under liquid N$_2$ into a Gatan Model 626 cryo-transfer holder, and subsequently into an FEI F20 TEM STEM instrument (FEI Company, Hillsboro, OR) equipped with a Gatan Tridium spectrometer (Gatan Inc., Pleasanton, CA) which was operated at 200 kV for the EELS experiments. Lamella thin-sections were similarly transferred into an aberration-corrected FEI Titan Themis S/TEM instrument (FEI Company, Hillsboro, OR) with a Gatan GIF Quantum 965 spectrometer (Gatan Inc., Pleasanton, CA) operated at 120 kV. During experiments on both instruments, the samples were maintained at approximately −180 °C.

ROI for high-resolution imaging and analysis were selected based on the sample position (over vacuum) and electron transparency. Given these criteria were met, the analysis was conducted on visible features with contrast differences and a detectable C signal. The approach to selecting the analysis region is described in detail in Supplementary Fig. 7 (analysis workflow).

Overview annular dark-field STEM images were obtained prior to the collection of EELS point and line spectra (wedge) and 2D maps (lamella). EELS data were collected with parameters to optimize EELS counts but minimize dose (Supplementary Table 4). For EELS maps collected on the Titan Themis instrument, DualEELS was used to simultaneously collect the low-loss and high-loss regions, and the zero-loss peak position was used to correct for energy shifts in the data due to large fields of view.

*Cryo-STEM-EELS spatial resolution:* Fundamentally, the spatial resolution of a measurement is limited by the STEM probe size, ranging from sub-Å (Titan instrument) to ~2 Å (F20 instrument). However, when performing EELS mapping over relatively large fields of view (in Figs. 3–5)—important for capturing relevant features in the soil specimen—the spatial sampling, i.e., the step size, sets the effective lower limit of spatial resolution relevant to statistical analysis of EELS data. In this study, step sizes used were an order of magnitude larger than the STEM probe size to minimize sample damage. The smallest step size used in this study was ~1 nm (10 Å), resulting in an effective spatial resolution of 2 nm set by the Nyquist limit (Figs. 3d, 4, and 5).

*Cryo-STEM-EELS damage assessment:* We assessed evidence of artifacts in the C K-edge resulting from the cryo-STEM-EELS analysis. For the organo–organic interface sample (Titan instrument), potential sample damage from cryo-STEM-EELS was assessed by repeated measurements of a semicrystalline iron oxide-

organic material serving as reference material for the volcanic semicrystalline soils analyzed. The reference material was prepared by precipitation of ferrihydrite (nominally $Fe_2O_3 \cdot 0.5H_2O$) in the presence of water-extractable organic matter (WEOM) derived from a soil organic horizon (Oa) at a 10:1 C:Fe ratio (WEOM extraction methods described in ref. [36]). Precipitation of ferrihydrite was completed using a low-concentration modification of standard laboratory ferrihydrite synthesis[69] and purified by dialysis (1000 D molecular weight cut-off). Ferrihydrite-WEOM suspensions were applied to Cu grids and air-dried. Samples were transferred at room temperature to an aberration-corrected FEI Titan Themis S/TEM instrument (FEI Company, Hillsboro, OR) with a Gatan GIF Quantum 965 spectrometer (Gatan Inc., Pleasanton, CA) operated at 120 kV. The samples were cooled after loading into the instrument for cryo-STEM-EELS to approximately −180 ℃.

Sample damage was assessed by 7 repeated EELS measurements (640 kx magnification) of the C K-edge over the same field of view ($1247 \times 1247$ Å$^2$). For the damage test, total dose ranged from ~10,000 (initial) to 67,000 (final) electrons (e$^-$) Å$^{-2}$ (Supplementary Table 4). Spectra were processed using the Cornell Spectrum Imager package[70] in ImageJ v. 2.0.0[71]. Background subtraction was performed using a standard linear combination of power laws (LCPL) with the background subtraction region of 209.9–259.3 eV. The C K-edge intensity was estimated by calculation of area under the curve (AUC) from 280.0 to 315.0 eV in the DescTools package for R in RStudio[72–74]. The ratio of the integrated area between lower (284.5–286.5 eV) higher-energy (286.5–289.0 eV) spectral regions (normalized to the total C area) was used to assess change in fine structure as a function of electron dose. For all AUC calculations, intensity counts less than 0 were excluded.

Only minor changes in total C intensity and low/high-energy ratios were observed across the damage test series (Supplementary Fig. 11). Both total C and low/high-energy ratios increased slightly between 10,000 and ~20,000 e$^-$ Å$^{-2}$ dose, suggesting that for the highest magnification EELS map (~11,000 e$^-$ Å$^{-2}$), the intensity of lower-energy features may be overestimated relative to lower-magnification EELS maps (Supplementary Fig. 11). However, the interpretations of shifts in fine structure ratios of the organo–organic interface are contained within the same EELS map. In addition, no appearance of new features with increasing doses (i.e., damage artifacts) were observed (Supplementary Fig. 11). The lack of artifact development attests to EELS data quality.

For the organo–mineral interface sample (F20 instrument), multiple point scans in the thin OM region adjacent to the organo–mineral interface (Fig. 2) were used to assess the effect of increasing dose (by a factor of 3.5) on the C K-edge EELS spectrum. Dose estimates are summarized in Supplementary Table 4. With increasing dose, a slight decrease in the ratio of high (region $x$)/low (region $y$) energy regions was observed (Fig. 2). For point 1, ratio $x/y = 1.1$ and for points 2 and 3, $x/y = 1.0$. However, the lack of change between points 2 and 3 despite a 1.25-factor increase in the dose (Supplementary Table 4) suggests that these differences may be due to fine-scale sample heterogeneity rather than beam damage. Further, the same dose was used for each point in the organo–mineral interface lines as for adjacent OM point 1, supporting the validity of comparing high-/low-energy ratios. Accounting for ranges in high-/low-energy features given potential sample damage, the enrichment of higher-energy C features at the organo–mineral interface (33%) could range from ~16 to 53% (minimum $x/y = 1.1$ vs. 1.3 and maximum $x/y = 1.0$ vs. 1.5) (Supplementary Table 2).

**Image and spectral analysis**. *Electron energy-loss spectra*: EELS data (spectrum images, line scans, and point data) were initially processed using the Cornell Spectrum Imager package[70] in ImageJ v. 2.0.0[71]. Background subtraction was performed using a linear combination of power laws (LCPL) function with the following background subtraction regions: C K-edge (210.0–259.5 eV), N K-edge (369.8–388.8 eV), and O K-edge (488.0–510.0 eV) for organo–organic interface spectra. For the organo–mineral interface sample, the 260.0 to 276.9–277.0 eV energy region was used for the C K-edge, and 369.0–378.8 to 385.4–394.2 eV for the N K-edge. The background subtraction was performed with 3-pixel over-sampling. One outlier point (at 405.8 eV) was excluded from all N K-edge spectra at the mineral interface due to detector error. For the EELS spectrum images and line scans, energy alignment was completed using the peak position of the zero-loss peak in low-loss EELS datasets paired to each measurement by nearby location or simultaneous measurement. For the adjacent OM to the organo–mineral interface, the high magnification precluded the use of zero-loss measurements and no energy shift was applied.

To estimate the intensity of non-normalized spectra for elemental and fine structural ratios, the integrated area was approximated by applying the AUC function (trapezoidal method) in the DescTools package for R in RStudio[72–74]. For all spectra, total C and N integration regions were set to 280.0–315.0 eV and 395.0–430.0 eV, respectively. For the organo–mineral interface, the higher- and lower-energy regions (called here regions x and y) were set to 286.6–289.0 eV and 284.0–286.5 eV, respectively. For the organo–organic interface, O integrated area was determined for 530.0–565.0 eV, and higher-energy (putatively alkyl C) and lower-energy (aromatic C) integrated areas were determined for 286.0–287.5 eV and 284.25–285.75 eV regions, respectively. For all AUC calculations, intensity counts less than 0 were excluded. The EELS spectra were not flattened or otherwise adjusted. For calculations of aromatic/alkyl ratios (e.g., Figs. 4 and 5) and other fine

structure analyses of C, the integrated area was normalized to the total C integrated area.

*EELS fine structure analysis*: Statistical analysis of the EELS fine structure for the organo–organic interface was conducted by MCR as described previously[37,75,76], performed in this study using Matlab v. R2017a. Briefly, MCR is an approach to decompose a spectrum image matrix into substituent components without a priori knowledge of chemical composition or use of analytical standards, given a set number of components and an initial selection of putative components. The resulting spectral components are then fit to the original data to produce maps showing component spatial distribution, such as those shown in Fig. 3.

## Data availability
The analytical microscopy data that support the findings of this study are available in the Cornell University eCommons Repository with the identifier https://doi.org/10.7298/6vtr-c668[77].

## Code availability
Code used for multivariate curve resolution (MCR) analysis is available on reasonable request to the authors.

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

## Acknowledgements

Funding for this study was provided by the NSF IGERT in Cross-Scale Biogeochemistry and Climate at Cornell University (NSF Award #1069193) and the Technical University of Munich Institute for Advanced Study. Additional research funds were provided by the Andrew W. Mellon Foundation and the Cornell College of Agriculture and Life Sciences Alumni Foundation. M.J.Z. and L.F.K. acknowledge support by the NSF (DMR-1654596) and Packard Foundation. This work made use of the Cornell Center for Materials Research Shared Facilities which are supported through the NSF MRSEC program (DMR-1719875). Additional support for the FIB/SEM cryo-stage and transfer system was provided by the Kavli Institute at Cornell (KIC) for Nanoscale Science and the Energy Materials Center at Cornell, DOE EFRC BES (DE-SC0001086). The FEI Titan Themis 300 was acquired through NSF-MRI-1429155, with additional support from Cornell University, the Weill Institute, and the KIC. The authors thank Katherine E. Grant and Louis A. Derry (Cornell University Earth and Atmospheric Sciences) for providing soil samples from the Polulu Flow, HI. The authors also thank John Grazul and Malcolm Thomas of the CCMR facility for technical assistance.

## Author contributions

A.R.P and J.L. developed the research question and scope and wrote the paper. D.A.M, L.F.K., B.D.A.L, and M.J.Z. provided guidance in method development and data interpretations. A.E. conducted sample preparation using the cryo-ultramicrotome and provided general technical assistance and method development. B.D.A.L., M.J.Z., and A.R.P. collected microscopy and spectroscopy data, conducted data analysis, and generated plots, figures, and images in the manuscript. All authors contributed to paper revisions.

## Competing interests

The authors declare no competing interests.
