## [Peer Review File · Nature Communications]

Reviewers' comments:

Reviewer #1 (Remarks to the Author):

The authors used novel methodological approaches to assess the nanometer-scale spatial properties of biogeochemical interfaces at mineral surfaces in soil. Knowledge about these spatial properties is key to better understand important ecosystem functions of soils such as nutrient recycling and climate change mitigation via long-term storage of organic carbon. Based on their data, the authors suggest a new "conceptual view" about the spatial structure of the biogeochemical interfaces.

In my opinion, the manuscript addresses key questions of recent research on soil organic matter, thus suits well to the scope of the journal and will be of wider interest in the field of biogeochemistry and environmental science. However, I think that before publication several revisions are needed, because (i) some arguments used to develop the new concept are not clearly enough outlined, and (ii) the fundamental mechanisms that could explain the findings of the authors are only vaguely addressed. Hence, I think the new concept can be enhanced before publication in order to be more helpful for the scientific community. I detail these shortcomings in my comments below.

First, the 'old view' of formation of multiple layers of organic molecules on mineral surfaces, as well as available experimental evidence supporting this old concept, are not clearly enough explained and discussed. The authors state that the old view of properties of organic matter coatings at mineral surfaces, as for instance proposed by Kleber et al. (2007), would address the micrometer scale (e.g., line 64, but also later in the discussion). However, the concept of zonal structure of coatings shown in Fig.2 of the article of Kleber et al. (2007) addresses the scale of a few molecules. Hence, the zones span rather a few nanometers. Also, the concept of Figure 2 in Kleber et al. (2007) does not clearly imply gradients of C forms (as stated by the authors, see Fig. 4 of the present manuscript); for instance, according to Kleber et al. (2007) carboxyl groups can accumulate directly at the surface as well as in more distance at the borders of hydrophobic zones and within the kinetic zone. Given this, I think that many parts of discussion about the old view should be more precise in order to demonstrate the real novel findings of the present work.

Second, the authors mention the role of mineral surfaces in organic carbon stabilization in several paragraphs, yet this topic is not well integrated in the argumentation. In particular, it is not discussed and clarified how the concept of the authors relates to the idea/concept of carbon stabilization by strong chemical bonds at biogeochemical interfaces which reduce bioavailability of the components (thus enhance their turnover times in soil). Is all of the organic matter bound in the biogeochemical interface stabilized, or only the molecules that are directly bound to the mineral surface? Moreover why should microbial residues be more strongly bound (thus stabilized?) within the interfaces than plant-derived molecules? Which chemical factors can explain the enrichment of N-rich components at certain zones of the interfaces? Can this really be deduced from variable chemical features between molecules, or do we additionally need to consider significant biological factors (for instance, microbial biofilm formation may reduce access of plant-derived DOM to mineral surfaces, etc.)? I am aware of that we still may not have enough empirical data to really answer such questions, but in my opinion the authors should tackle them more in the discussion and provide their views even if these are hypothetical in nature. This would be needed to clarify the mechanistic fundament of the new concept proposed by the authors (in particular the part where the results on chemical functional groups is presented is largely descriptive, and does not address mechanisms of chemical interactions between molecules). Note that the model of Kleber is more elaborate on this aspect, as it proposes zones with strong bonds, where molecules might be more stabilized, and a 'kinetic' zone with less stabilized components.

Reviewer #2 (Remarks to the Author):

This is a well executed and thoughtful paper which highlights a need to further explore the role N containing compounds in soil C sequestration. The authors have adequately demonstrated that N containing compounds such as microbial amino acids can bind to organics creating layers of organo-organic complex that are contributing to protection of soil C.

I hope that the authors will encourage similar experiments with other soil types providing information is necessary to establish a more complete picture of SOM, SOM formation and C sequestration mechanisms across scale in a range of soil ecosystems.

One minor issue is that an acronym had been introduced but hadn't been described/spelled out. I'm not entirely sure that any audience would know what NEXAFS is. It may be worth it to go through the manuscript once more to ensure that abbreviated techniques have been spelled out before using the acronym.

Reviewer #3 (Remarks to the Author):

So, this is an interesting manuscript that uses state-of-the-art technology to look at organic carbon and organic mineral complexes in situ. It purports to show what the authors consider new observations of disorder and heterogeneity in these complexes. The authors state that previous work hypothesised and/or showed an order (they initially call it "layering") to these complexes.

There are quite a few problems with the manuscript however:

[1] Despite the claims of the authors there is no viable model or indeed dataset that supports or claims order or layering in any of the properties that they mention. Patchiness is an inherent property in the majority of soils, especially at high spatial resolutions – interestingly probably less so in the soil that they choose. The data of Nunan et al, using thin sections, as the authors here do, shows this with a focus on deep statistical quantification of microbes in soil at the micron scale and above. Indeed, unlike the authors, the Nunan suite of papers actually quantify disorder, or statistical heterogeneity. Now this is on two aspects of soil – physical structure and microbial numbers and location. But we all know these map very well onto soil-C. So, I do not have any substantive issues with the data in the authors papers (see queries below), just the fact that they seem surprised by their own data. Soil is the most complex biomaterial in the world and spatial and temporal heterogeneity, in form and function (hot spots and hot moments), is a given. There are several other authoritative papers who have also now shown that heterogeneity is a key characteristic in soils and such layering as the authors propose is simply not typically present. Where it could be present in the field, is when org-C is incorporated into soil. This would be a layering of sort but soon, after chemical and microbial degradation, significant heterogeneity (disorder) will be induced. Tushimoto Hattori did have a concept of inner and outer bacterial populations in individual aggregates. However, this fell by the wayside. The concept of order/disorder in soils is taken up nicely by the self-organised model which has been proven under some conditions by several groups. So, my main issue is with the unstated hypothesis of layering/order. Even if the authors are correct, I still do not see any substantially new data. We all know already the import of N in C-sequestration. The literature is replete with that data. Knowing that it occurs at small scales is obvious as it is manifest at larger scales. They state that "The existence of organic patches and organo-organic interfaces with different C composition than those at organo-mineral interfaces generates important information for sequestering soil carbon and predicting its changes." We know much already about patchiness. This manuscript does however, emphasise some neat technology to further interrogate the soil ecosystem. But it does not offer any new insights.

[2] The procedures adopted are clearly state of the art. I am not experienced in these procedures and have a few queries borne out of ignorance. What is the calibration involved in these systems? Are the inferences from peaks (form) always consistent to specific chemical constructs (function)? Also, the

images seem poor. But is that typical for these systems – it looks like gradients of greyscale with sometimes very diffuse boundaries. How nano is it really?

[3] Choice of soil. The ..'soil samples were collected by horizon quantitatively from sub-soils from the Pololu Flow on Kohala, HI with approximately 350,000 years of soil development." The choice seems quite bizarre as the authors are trying to make a statement about the typical composition and layering of soil complexes. What does "collected by horizon quantitatively" mean? Also, why a subsoil? Also, what was the sampling procedure? Given that spatial and temporal heterogeneity is key to the form and function of soil systems, and indeed the authors layering idea, this is a critical area. Why choose such a restrictive size of soil? Storage of soils seems suspect as well: "Soils were stored at field moisture conditions at 9°C prior to cryo-STEM sample preparation and imaging". Why choose such a storage regime? What were the field moisture conditions and why 9°C, and for how long? Again, these are important areas as soils conditions chemical, physical and biological, potentially could change significantly in such storage conditions. Was the soil sampled undisturbed? This area seems very poor and poorly thought out.

[4] What was the methodology of the quantification.? There seems to be almost zero info about this. What was the variability in the measurements? Where are the confidence intervals? What was the sampling procedure on the thin sections – I note some info but not much and certainly not about the placement of the measuring grids.

Summary

This paper is very much focussed on a range of techniques that potentially could provide new insights to soil at the nano scale. However, it fails on a number of important points from the original, unstated, hypothesis through to sampling and onto analysis. I sense that it's from a PhD that focusses on techniques rather than quantification. My recommendation is "Reject".

Response to Review

Editor's Comments:

Dear Dr Lehmann,

First, please accept my sincerest apologies for the delay in reaching an initial decision on your manuscript. Unfortunately, one of the secured referees disappeared after being granted an extension, and then with the pandemic it was difficult to secure a replacement who could turn around a report over a tight timeline. Nevertheless, your manuscript entitled "Organo-organic and organo-mineral interfaces in soil at the nanometer scale" has now been seen by 3 referees, whose comments are appended below. You will see from their comments copied below that while they find your work of considerable potential interest, they have raised quite substantial concerns that must be addressed.

Reviewer #1 noted that the mechanisms underlying your observations needed to be better explained, and contrasted more substantially with previous conclusions about these dynamics that your results cast in a new light. Reviewer #3 voiced concern about hypotheses related to layering in soil aggregates, questioned whether the choice of soil samples were applicable to other systems, and felt that the methods required more detail. Overall, there was a general consensus that the novelty and advance of your work needed to be made much more clear before the paper can be reconsidered. In light of these comments, we cannot accept the manuscript for publication, but would be interested in considering a revised version that addresses these serious concerns.

Response: Thank you. In the revision, we addressed the questions and comments raised by the reviewers through several major conceptual additions and clarifications in the text. Major changes (outlined in detail for each reviewer) include:

- (1) In a revised Figure 1, we clarified how our very fine-scale observations of the interfaces between organic and mineral phases in soil refines our understanding of the hierarchy of organic matter (OM) spatial heterogeneity, in the context of previous observations at the scale of microaggregate cross-sections (showing inherent heterogeneity of OM and soil microstructures) and organic coatings within organo-mineral assemblages (showing layered accumulation of OM);
- (2) Throughout the manuscript, we expanded on and clarified how our data contributes to better understanding of nitrogen (N)-rich OM interaction with reactive iron (Fe) and aluminum (Al) minerals, and generates novel insights into how these mechanisms might differ between organo-mineral and organo-organic interactions;
- (3) In the Methods section, we better explained the selection of soil type, depth, and analysis approach, and provided a new analysis pipeline (also through a Supplementary Figure) to better communicate considerations unique to high-resolution imaging and spectroscopy techniques; and
- (4) We included a new Figure 5 in the main manuscript (based on a former Supplementary Figure), with additional data on the C/N and C/O ratio trends

across the organo-organic interface at 2-nm lateral resolution. This figure adds additional support to observations of ordered gradients in OM composition at <5 nm scales, enabled by an order of magnitude improvement in spatial resolution compared to previous imaging and spectroscopy of soil samples.

Please note that line numbers refer to the revision without tracked changes.

Reviewers' comments:

Reviewer #1 (Remarks to the Author):

The authors used novel methodological approaches to assess the nanometer-scale spatial properties of biogeochemical interfaces at mineral surfaces in soil. Knowledge about these spatial properties is key to better understand important ecosystem functions of soils such as nutrient recycling and climate change mitigation via long-term storage of organic carbon. Based on their data, the authors suggest a new "conceptual view" about the spatial structure of the biogeochemical interfaces.

In my opinion, the manuscript addresses key questions of recent research on soil organic matter, thus suits well to the scope of the journal and will be of wider interest in the field of biogeochemistry and environmental science. However, I think that before publication several revisions are needed, because (i) some arguments used to develop the new concept are not clearly enough outlined, and (ii) the fundamental mechanisms that could explain the findings of the authors are only vaguely addressed. Hence, I think the new concept can be enhanced before publication in order to be more helpful for the scientific community. I detail these shortcomings in my comments below.

First, the `old view` of formation of multiple layers of organic molecules on mineral surfaces, as well as available experimental evidence supporting this old concept, are not clearly enough explained and discussed. The authors state that the old view of properties of organic matter coatings at mineral surfaces, as for instance proposed by Kleber et al. (2007), would address the micrometer scale (e.g., line 64, but also later in the discussion). However, the concept of zonal structure of coatings shown in Fig.2 of the article of Kleber et al. (2007) addresses the scale of a few molecules. Hence, the zones span rather a few nanometers. Also, the concept of Figure 2 in Kleber et al. (2007) does not clearly imply gradients of C forms (as stated by the authors, see Fig. 4 of the present manuscript); for instance, according to Kleber et al. (2007) carboxyl groups can accumulate directly at the surface as well as in more distance at the borders of hydrophobic zones and within the kinetic zone. Given this, I think that many parts of discussion about the old view should be more precise in order to demonstrate the real novel findings of the present work.

Response: Thank you for highlighting that the spatial arrangement or architecture of OM accumulation on mineral surfaces (e.g., accumulation in organized layers) is conceptually distinct from the scale of OM accumulation on mineral surfaces (e.g., the thickness of layers). We agree that the two were not clearly distinguished in the original manuscript, and we have made substantial revisions in the text to refocus the

conceptual framework presented. In the revision, we highlight the contrast between our observations of disordered distribution of OM features 0.3-1 micrometer (μm) in size to the previous observations of 1-10 μm -thick layers of OM accumulation on mineral surfaces using other techniques (summarized in Response Figure 1 and in the revised Figure 1). We also highlight the known patchiness of OM distribution at the scale of a soil aggregate cross-sections (lines 51-53 and in Figure 1), in contrast to organic coatings on mineral particles at finer scales. Previous observations (e.g., with X-ray microscopy) have shown ordered layers of OM composition, but using a cryo-electron microscopy approach which affords an order of magnitude improvement in spatial resolution, we show disordered, patchy and chemically (i.e., functional group composition) distinct structures (0.3-1 μm in size) within such mineral coatings. Furthermore, at a very fine scale, we show evidence for <5 nm-thick layers of identifiable OM forms. The high spatial resolution (down to single-digit nm) approaches a more meaningful (though still larger than the size of individual molecules) scale to apply the Kleber et al. (2007) model. At this scale, our observations do not contradict the discrete “zones” (i.e., layers of OM composition) of the Kleber model, but we highlight that zonal arrangement in space may occur not only in the context of mineral surfaces, but also at the boundaries of irregularly shaped organic patches within a matrix of patchy and distinct organic matter forms that were not distinguishable within the interfaces that have previously been imaged (such as in Lehmann and Solomon 2010, using NXAFS-STXM). This distinction is highlighted in the revised manuscript Figure 1 and in the text in lines 233-250.

Second, the authors mention the role of mineral surfaces in organic carbon stabilization in several paragraphs, yet this topic is not well integrated in the argumentation. In particular, it is not discussed and clarified how the concept of the authors relates to the idea/concept of carbon stabilization by strong chemical bonds at biogeochemical interfaces which reduce bioavailability of the components (thus enhance their turnover times in soil).

1. Is all of the organic matter bound in the biogeochemical interface stabilized, or only the molecules that are directly bound to the mineral surface?

Response: Thank you for raising a thought-provoking question, which we feel provides an important direction for future work and was incorporated into the manuscript as a comment on the link between our observations and limitations of inferring “persistence” (lines 288-296). The association between reactive Fe and Al mineral abundance and increased long-term stability of bulk SOM has been made for decades, often attributed to the formation of chemically-stable inner-sphere type covalent bonding interactions (e.g., Torn et al., 1997, Torn et al., 2009; Kramer et al., 2012; Kleber et al., 2015), with the inference that mineral bonding limits microbial accessibility for decomposition on the substrate (e.g., discussed in Deng and Dixon, 2002), and lowers reversibility of bonded OM substrates back to the soil solution (i.e., slowly reversible adsorption) (now stated in lines 288-290). For OM not co-located with minerals, no conferral of mineral protection via direct interactions can be assumed. However, the conferral of persistence (e.g., C turnover time via $\Delta^{14}\text{C}$) as a function of distance to the mineral surface has not been demonstrated with direct imaging and spectroscopic approaches, to our knowledge. In

our study, we cannot resolve the age of the OM observed, aside from presumed long residence times from bulk soil measurements (often greater than 8,000 radiocarbon years in soils of this system). In the literature, sequential removal of OM associated with minerals by repeated dissolution points to higher extractability (i.e., corresponding to less persistence) of outer layers of OM relative to that bonded directly to the mineral surface (Coward et al., 2018). However, to demonstrate a difference in persistence between OM at the mineral surface compared to at a further distance requires the methodological challenge of measurements that could pair high-resolution imaging with quantification of C composition and natural abundance radiocarbon (^{14}C) measurements: this may be a fruitful area for future study, noted in **lines 293-296** (e.g., with advancement of monochromated EELS techniques, discussed in Jakisaari et al., 2018, and Hachtel et al., 2019).

2. Moreover why should microbial residues be more strongly bound (thus stabilized?) within the interfaces than plant-derived molecules? Which chemical factors can explain the enrichment of N-rich components at certain zones of the interfaces?

Response: Excellent question: we had not sufficiently explained this concept in the text and have included the concept more visibly in the revision **(lines 57-64, 123-142)**. The basic principle is outlined in Lehmann and Kleber (2015 *Nature* 528, 60-68; figure reproduced below), by reviewing the literature and observing that greater decomposition of plant/animal residues by microorganisms will make molecules smaller and more functionalized (mainly oxidation and O rich, but also more N rich), which increases likelihood of interaction with minerals (in the paper called the “Soil Continuum Model”). We have now added more information about the reactivity of N-rich decomposition products into the introduction **(lines 79-81)**.

Figure taken from Lehmann J and Kleber M 2015 The contentious nature of soil organic matter. Nature 528, 60-68.

3. Can this really be deduced from variable chemical features between molecules, or do we additionally need to consider significant biological factors (for instance, microbial biofilm formation may reduce access of plant-derived DOM to mineral surfaces, etc.)?

I am aware of that we still may not have enough empirical data to really answer such questions, but in my opinion the authors should tackle them more in the discussion and provide their views even if these are hypothetical in nature. This would be needed to clarify the mechanistic fundament of the new concept proposed by the authors (in particular the part where the results on chemical functional groups is presented is largely descriptive, and does not address mechanisms of chemical interactions between molecules). Note that the model of Kleber is more elaborate on this aspect, as it proposes zones with strong bonds, where molecules might be more stabilized, and a `kinetic` zone with less stabilized components.

Response: Thank you for pointing out this important distinction between the observed OM chemical composition and its expected source (i.e., microbially processed or intact plant residue). The nature of chemical bond stability, as such, is likely not categorically

different between OM source: i.e., a carboxylic group ligand exchange reaction with a hydroxide mineral functional group is chemically identical, regardless of its original source. In our study, the enrichment of N and the presence of N-substituted carboxylic acids together suggest microbial origin and protein (amide)-mineral interactions, which have been linked to OM persistence by a number of association mechanisms, including phosphate group bonding, hydrogen bonding, and electrostatic interactions (Knicker et al., 2011; Keiluweit et al., 2012). Higher relative amino acid and protein content is expected for residues that have been cycled through microbial biomass, rather than for intact plant residues; consequently, protein-mineral interactions are often associated with microbial origin. However, in our study, we cannot verify the source (i.e., microbial or unprocessed plant residues) of the compounds detected, so we have presented the results in a context more constrained to the specific OM identified by EELS (N-rich oxidized OM at the mineral surface, N-rich alkyl OM at the organo-organic interface). We make this clearer in the revision (lines 111-121) and encouraged by this referee (since we were indeed more cautious about the implications of our results than we needed to be), present several possible sources and links to broader scale concepts of OM stabilization. In this revision, we made sure that the concept of microbial residue contribution to persistent carbon is included as an emerging stabilization paradigm. The concepts of microbial cell constituents accumulation, microbial colonization of mineral surfaces and residue co-location, microbial “signature” (e.g., lower C/N, lower lignin content, and stable isotope composition) of dense (mineral) fractions, and correlations between SOM properties and microbial community characteristics (e.g., substrate use efficiency) (e.g., discussed in Sollins et al., 2006; Miltner et al., 2011; Kallenbalch et al., 2016) are consistent with our results of functional group composition obtained at single-digit nm resolution. Collectively, these studies have been used to develop microbial SOM formation frameworks (e.g., the Microbial Efficiency-Matrix Stabilization, or MEMS framework) (Cotrofu et al., 2013), which have now been incorporated into the text in lines 136-142.

Reviewer #2 (Remarks to the Author):

This is a well executed and thoughtful paper which highlights a need to further explore the role N containing compounds in soil C sequestration. The authors have adequately demonstrated that N containing compounds such as microbial amino acids can bind to organics creating layers of organo- organic complex that are contributing to protection of soil C.

I hope that the authors will encourage similar experiments with other soil types providing information is necessary to establish a more complete picture of SOM, SOM formation and C sequestration mechanisms across scale in a range of soil ecosystems.

One minor issue is that an acronym had been introduced but hadn't been described/spelled out. Im not entirely sure that any audience would know what NEXAFS is. It may be worth it to go through the manuscript once more to ensure that abbreviated techniques have been spelled out before using the acronym.

Response: Thank you for the suggestion: all instances where abbreviations were used were checked and defined in the revision.

Reviewer #3 (Remarks to the Author):

So, this is an interesting manuscript that uses state-of-the-art technology to look at organic carbon and organic mineral complexes in situ. It purports to show what the authors consider new observations of disorder and heterogeneity in these complexes. The author state that previous work hypothesised and/or showed an order (they initially call it "layering") to these complexes.

There are quite a few problems with the manuscript however:

[1a] Despite the claims of the authors there is no viable model or indeed dataset that supports or claims order or layering in any of the properties that they mention.

Patchiness is an inherent property in the majority of soils, especially at high spatial resolutions - interestingly probably less so in the soil that they choose. The data of Nunan et al, using thin sections, as the authors here do, shows this with a focus on deep statistical quantification of microbes in soil at the micron scale and above. Indeed, unlike the authors, the Nunan suite of papers actually quantify disorder, or statistical heterogeneity. Now this is on two aspects of soil - physical structure and microbial numbers and location. But we all know these map very well onto soil-C. So, I do not have any substantive issues with the data in the authors papers (see queries below), just the fact that they seem surprised by their own data. Soil is the most complex biomaterial in the world and spatial and temporal heterogeneity, in form and function (hot spots and hot moments), is a given. There are several other authoritative papers who have also now shown that heterogeneity is a key characteristic in soils and such layering as the authors propose is simply not typically present.

Where it could be present in the field, is when org-C is incorporated into soil. This would be a layering of sort but soon, after chemical and microbial degradation, significant heterogeneity (disorder) will be induced. Tushimoto Hattori did have a concept of inner and outer bacterial populations in individual aggregates. However, this fell by the wayside. The concept of order/disorder in soils is taken up nicely by the self-organised model which has been proven under some conditions by several groups. So, my main issue is with the unstated hypothesis of layering/order. Even if the authors are correct, I still do not see any substantially new data.

Response: Thank you for highlighting this point of the known high heterogeneity of soil and its organic matter. In our original submission we had failed to make it sufficiently clear that we are observing the spatial arrangement in soil at 1-4 orders of magnitude greater resolution than previous studies using scanning transmission X-ray microscopy (STXM) with near-edge X-ray absorption fine structure (NEXAFS) (about 30-50 nm resolution) and Fourier-transform infrared (FTIR) microscopy (about 5-10 μm resolution) (Lehmann and Solomon, 2010). In the revision, we therefore address an important gap in the discussion of our observations in the context of measurements at

the scale of soil aggregate microstructure and microbial population distribution. For a summary of this revision, please see the new conceptual sketch below (Response Figure 1) and following discussion.

References: 1. Nunan et al. 2001; 2. Raynaud and Nunan 2014; 3. Lehmann et al. 2007; 4. Steffens et al., 2017; 5. Lehmann et al., 2008; 6. Kinyangi et al., 2006; 7. Solomon et al., 2012; 8. Milne et al., 2011; 9. Lehmann and Solomon, 2010.

Response Figure 1. Hierarchy of scale in direct imaging paired with spatial spectroscopic characterization of organic matter composition.

To better inform the discussion of scale and resolution, here we define “scale” as the typical field-of-view of an image collected, “resolution” as the minimum distance over which a technique can extract statistically significant distinct compositional information from the material of interest (e.g., by X-ray absorption, electron energy loss, or Fourier transform infrared spectral data), and “size” as the physical dimensions of a feature (e.g., 0.3-1 μm -size patches of distinct organic matter composition). In Response Figure 1, we highlight the distinction among: (A) previous observations of heterogeneity of aggregate mineral structures, pore space, microbial habitat and distribution, and organic matter distribution, (B) previous observations of organo-mineral assemblages in soil aggregate thin sections, showing relatively homogeneous “layering” of mineral surface coatings, and (C) the current observations. Our new observations of organic functional group composition, well below the resolution limits of previously used techniques (1-4 orders of magnitude), revealed heterogeneous OM structures with gradients in OM composition at their edges (occurring at the 0.005 to 0.01- μm scale) and provided direct evidence for the enrichment of N at both mineral and organo-organic interfaces.

As noted by the reviewer, the heterogeneous distribution of OM, microorganisms, and physical structures at the aggregate cross-section scale has been frequently assessed by combining imaging and spectroscopy measurements with roughly $\sim 0.2\text{-}\mu\text{m}$ resolution, using techniques such as scanning transmission X-ray microscopy (STXM) paired with near-edge X-ray absorption fine structure (NEXAFS) spectroscopy (shown here in **A**). A number of previous measurements with other techniques also support the existence of heterogeneity at the scale of aggregates (as the referee correctly noted with key references), including optical microscopy, scanning electron microscopy, and nanoscale secondary ion mass spectrometry. Despite the well-established existence of complexity and heterogeneity at the aggregate cross-section scale, measurements focused directly on organo-mineral assemblages (using $\sim 0.05\ \mu\text{m}$ resolution) have previously shown relatively homogeneous distribution in discrete layers of organic matter composition (e.g., carboxylic, aliphatic, aromatic functional groups) on mineral coatings, shown here in **B**. These observations (previously published) form the underpinning of our discussion of the “layering” concept.

The hierarchy of scale discussed above is now highlighted in the revised manuscript in Figure 1. While our observations of discrete patches of organic composition (manuscript Fig. 3) reflect the concept of patchiness at aggregate cross-section scales, we argue that the existence of irregular OM structures ($0.3\text{-}1\ \mu\text{m}$ in size) at the scale of the organo-mineral or organo-organic interface cannot be automatically inferred from previous observations at intermediate scales (such as Lehmann and Solomon, 2010, also shown in the Response Figure 1, above).

Additionally, our ability to resolve different carbon composition within and immediately adjacent to $\sim 1\ \mu\text{m}$ -size features pointed towards the existence of relatively discrete “layers” of organic matter composition, shown to be up to $<5\ \text{nm}$ ($0.005\ \mu\text{m}$) in thickness (new Fig. 5 in the manuscript, based on a previous Supplementary figure). We added additional support for these observations by calculating new C/N and C/O ratio data across the interface using 2-nm lateral resolution, in addition to measurements of aromatic/alkyl ratio. These observations point to a divergence from the concept of patchiness when viewed at ultrafine resolution, approaching a scale with relevance to models of interfacial interactions and adsorption (e.g., the “zonal” model in Kleber et al., 2007). In summary, in the revision, we have expanded the discussion of the order of magnitude difference in scale considered here relative to observations of patchiness at the aggregate cross-section scale and larger,- highlighting that some properties (e.g., $<5\ \text{nm}$ thick “layers” of organic matter composition between organic phases) are in contrast to larger-scale observations.

[1b] We all know already the import of N in C-sequestration. The literature is replete with that data. Knowing that it occurs at small scales is obvious as it is manifest at larger scales. They state that "The existence of organic patches and organo-organic interfaces with different C composition than those at organo-mineral interfaces generates important information for sequestering soil carbon and predicting its changes." We know much already about patchiness. This manuscript does however, emphasise some neat technology to further interrogate the soil ecosystem. But it does not offer any new insights.

Response: Thank you for raising an important question about the novelty of our work specific to the importance of N. In the revision (lines 123-142), we incorporate additional support for the indirect association of N-rich OM with reactive metals (primarily Fe) (e.g., Knicker et al., 2011; Keiluweit et al., 2012; Heckman et al., 2018), including discussion of interaction mechanisms that have been hypothesized (Knicker et al., 2011; Keiluweit et al., 2012) or demonstrated in model systems (using atomic force microscopy, in Newcomb et al., 2017). Our observations provide, to our knowledge, the first direct imaging and spectroscopic evidence at a resolution (up to single-digit nm) appropriate to unambiguously detect N-substituted carboxylic or N-rich alkyl OM chemical composition at mineral or organic phase interface, respectively, providing evidence for both co-location and mechanistic inference in a natural soil sample. The revised manuscript addresses the referee's concern and makes this much clearer (including through the revised figure).

Additionally, while the body of literature regarding N-rich OM enrichment in mineral fractions (e.g., soil density fractions, <2 μm size fractions, and selectively extracted pools) is not specific to reactive metals (N-rich material may also accrue without any interactions with minerals), much of the work to evaluate mechanisms of actual interactions is focused on OM interactions with reactive Fe (e.g., Keiluweit et al., 2012; Newcomb et al., 2017). Here, we highlight observations that provide new insight into N involvement in OM interactions with Al ("organo-mineral") (in lines 151-155) and OM interactions with different OM phases ("organo-organic"). While assuming similarity between OM-Al and OM-Fe interactions is theoretically sound (yet benefits from the experimental proof provided here), the mechanisms of N involvement in organo-organic interactions in natural soils are largely unknown, and are likely distinct from OM-metal interactions. Our observations offer a first look at these features and a potential impetus for further work (highlighted in lines 144-155). We make this clearer in the revision.

At a broader scale, we highlight the emerging consensus that N promotes organic matter accumulation beyond the stoichiometric need of microorganisms, but that major Earth systems models (e.g., CLM-CN 5.0) treat soil N availability in a classical framework as a potential limitation on SOM decomposition rates (e.g., low C/N increases decomposition and N release) (Lawrence et al., 2018). Further, from landscape or ecosystem scale experiments using N additions to soil, long-term N fertilization has shown mixed results: for instance, work from Riggs et al. (2015) highlighted decreased SOM decomposition rates and facilitation of occlusion under N fertilization, but recent work from Kazanski et al. (2019) showed no or inconsistent effects of N addition and cautioned against broad generalizations of SOC accrual effects after N additions. Clearly, the underlying mechanisms of the effect of N on soil organic carbon accrual is not yet fully understood. Further, the distinction between SOM accumulation (i.e., amount) and persistence (i.e., turnover time) with respect to N involvement in stabilization is not well known. The potential for system-specific (i.e., dependent on soil physicochemical properties) N effects on SOM accumulation/decomposition proposed by Kazanski et al (2019) emphasizes the need to interrogate the biophysical mechanism of SOM persistence that may be conferred by N,- such as through surface interactions, for which our study provides clear incentive. The text above was incorporated into the revision in lines 261-275.

[2] The procedures adopted are clearly state of the art. I am not experienced in these procedures and have a few queries borne out of ignorance. What is the calibration involved in these systems? Are the inferals from peaks (form) always consistent to specific chemical constructs (function)? Also, the images seem poor. But is that typical for these systems - it looks like gradients of greyscale with sometimes very diffuse boundaries. How nano is it really?

Response: A discussion of the calibration, interpretation of spectral features, and the assignment of lower limits to spectral and spatial resolution is presented in Response Figure 2. Thank you for suggesting that we further explain the true “nanoscale” nature of this technique. While the STEM-EELS technique in general is capable of up to sub-angstrom (<0.1 nm) spatial resolution paired with the ability to distinguish chemical structure, there are several factors that will affect the spatial resolution of a particular measurement, including (1) the optical setup of the electron microscope (probe size), (2) the analysis objective (tradeoff between field of view and resolution set by the sampling rate), (3) the signal/noise ratio (set by the tolerable elctron dose as well as the C amount), (4) optimization of spectral vs. spatial resolution, and (5) the ability to resolve interpretable spectroscopy data. In the methods section, we now define more explicitly the spatial resolution achieved here (lines 452-460): “Fundamentally, the spatial resolution of a measurement is limited by the STEM probe size, ranging from sub-Å (Titan instrument) to ~2Å (F20 instrument). However, when performing EELS mapping over relatively large fields of view (in Figs 3-5) – important for capturing relevant features in the soil specimen – the spatial sampling, i.e., the step size, sets the effective lower limit of spatial resolution relevant to statistical analysis of EELS data. In this study, step sizes used were an order of magnitude larger than the STEM probe size to minimize sample damage. The smallest step size used in this study as ~1 nm (10 Å), resulting in an effective spatial resolution of 2 nm set by the Nyquist limit (Fig. 3D, Figs 4 and 5).”

While the 1-nm step size set the lower limit of resolution, various step sizes (i.e., pixel size in EELS maps) were selected as appropriate for the scale of observations while also minimizing sample damage: e.g., to describe the overall shape of organic features (Figure 3C), ~15 nm step size was used, while analysis of the interface (Figure 3D) used ~1 nm step size.

1. Calibration

Microscope alignment performed with each use (standard operating procedures)
Spectrum energy position correction and energy resolution: Zero loss peak position and width

2a. EELS core loss spectrum features

Electron energy loss measures the energy of electrons that interact with a material and scatter (inelastically) at energies corresponding to element binding energies (K-shell for C and N)

2b. Statistical analysis of EELS maps

Component spectra with unique features (i.e., not spectrum noise) are identified within an EELS map using multivariate curve resolution (MCR). Energy positions of features in the MCR spectra can therefore be assigned to known bond excitation energies and mapped spatially.

3. Spatial resolution

We were able to resolve distinct carbon composition in damage-sensitive SOM to single digit-nanometer resolution, roughly an order of magnitude higher than analogous X-ray imaging-spectroscopy techniques (e.g., scanning transmission X-ray microscopy with near-edge X-ray absorption fine structure, STXM-NEXAFS)

Response Figure 2. Description of scanning transmission electron microscopy (STEM) with electron energy loss spectroscopy (EELS) calibration, interpretation, and spatial resolution.

[3] Choice of soil. The ..'soil samples were collected by horizon quantitatively from sub-soils from the Pololu Flow on Kohala, HI with approximately 350,000 years of soil development." The choice seems quite bizarre as the authors are trying to make a statement about the typical composition and layering of soil complexes. What does

"collected by horizon quantitatively" mean? Also, why a subsoil? Also, what was the sampling procedure? Given that spatial and temporal heterogeneity is key to the form and function of soil systems, and indeed the authors layering idea, this is a critical area. Why choose such a restrictive size of soil? Storage of soils seems suspect as well: "Soils were stored at field moisture conditions at 9°C prior to cryo-STEM sample preparation and imaging". Why choose such a storage regime? What were the field moisture conditions and why 9°C, and for how long? Again, these are important areas as soils conditions chemical, physical and biological, potentially could change significantly in such storage conditions. Was the soil sampled undisturbed? This area seems very poor and poorly thought out.

How do the soils samples address typical composition and layering of soil complexes?

Response: While Andisols occupy a relatively small total land area globally (0.7%), this soil type stores a disproportionate amount of SOC (1.3% of global SOC to a depth of 1 m) (Eswaran et al., 2000), and have been studied as an archetype of reactive Fe and Al control on SOM persistence for several decades. The most persistent SOM in this study system is expected to be found in soils after between 20,000 and 1,500,000 years of soil development (Torn et al., 1997), which was our rationale for selecting this 350,000 year-old site (expanded in the Methods Section 2). Andisols are the most likely soil type to contain spatially ubiquitous examples of OM-reactive metal interfaces for evaluation, an essential consideration for low-throughput but very high-resolution techniques employed in this study. We acknowledge that Andisols represent the endmember of short-range ordered (SRO), highly reactive Fe and Al mineralogy. In the revision, we clarify that our data do not indicate the abundance of the observed interfaces and that additional data from other soils are desirable (lines 297-299); however, reactive metal mineral phases are broadly distributed in most soil types as surface coatings on silicate clays and primary minerals, Fe precipitates and OM co-precipitates under fluctuating redox conditions, and SRO minerals in low-pH forest soils (e.g., Spodosols) (noted in the text in lines 123-136). The abundance of reactive Fe and Al phases (generally higher in lower pH systems) is considered an important predictor of SOM contents at the continental scale (Rasmussen et al., 2018). We made this clearer in the revision.

What does "collected by horizon quantitatively" mean?

Response: This term refers to collection of soil from the top to bottom edges of a soil horizon in order to represent the average bulk composition of a horizon, rather than collection of specific depth increments. Genetic horizon boundaries were determined in the field using transitions in structure, texture, and color. This clarification is added in the Methods Section 2.

Why a subsoil?

Response: In this study, the aim was to isolate samples with the greatest abundance of mineral-associated SOM and analyze the organo-mineral interface at a very small scale. With depth, the proportion of mineral-associated SOM is expected to increase, and the bulk SOM age is generally older, suggesting a greater contribution of mineral-

mediated SOM stabilization (Kleber et al., 2005; Schmidt et al., 2011). In the revision, we highlight the expected shift towards lower particulate and higher mineral-associated SOM in subsoil in the Methods Section 2. Since our study focuses on organo-mineral interactions, we aimed to limit the influence of more recent plant particulate OM and contribution of vegetation or land-use influence on SOM composition, which decreases with depth generally and specifically in our specific study system (Kelly et al., 1998; Chadwick et al., 2007; Inagaki et al., 2020).

What was the sampling procedure? Given that spatial and temporal heterogeneity is key to the form and function of soil systems, and indeed the authors layering idea, this is a critical area. Why choose such a restrictive size of soil?

Response: In the revision, the sampling procedure with respect to field soil collection was expanded on in Methods Section 2. Soil pits were excavated by hand to ~ 1 m and sampled for genetic horizon using approaches described in Kramer et al. (2012) and Grant et al. (2019). Distinct genetic horizons were defined using standard characteristics outlined in the US Soil Survey (Schoeneberger et al., 1988) and named by depth increment (both horizons 0.70-0.90 m). Sampling procedure with respect to aggregate selection for cryo-sectioning and microscopy was expanded on in **lines 375-393**. Subsamples (~100 g) of bulk soils from two subsoil horizons were gently wet-sieved without added pressure to <150 µm diameter to maintain natural microaggregates. This size fraction was selected because microaggregates are expected to represent the aggregate fraction with greatest contribution of persistent carbon (Edwards and Bremner, 1967; Tisdall and Oades, 1982; Six et al., 2000). Approximately 0.25 g of sieved aggregates were slowly hydrated using the method described in **lines 380-383**, and one aggregate per soil horizon was selected manually to conduct cryosectioning and FIB milling. Because our aim was to directly analyze characteristics of organo-mineral and organo-organic interfaces at a very high spatial resolution, we did not consider other aspects of spatial heterogeneity at increasingly larger scales here, such as distribution of particulate organic matter, microbial populations, pore distribution, root structure, or intra- or inter-horizon spatial distribution (all several orders of magnitude larger). For these important considerations operating at much larger scale, we now incorporate the body of literature that emphasizes the inherent heterogeneity of soils and specifically at the aggregate thin section scale (discussed above in Response Figure 1 and in the revised Fig. 1).

Storage of soils seems suspect as well: "Soils were stored at field moisture conditions at 9°C prior to cryo-STEM sample preparation and imaging". Why choose such a storage regime? What were the field moisture conditions and why 9°C, and for how long? Again, these are important areas as soils conditions chemical, physical and biological, potentially could change significantly in such storage conditions.

Response: Expansion on methods with respect to storage is now discussed in the text in **lines 339-343**. Soils were kept in coolers immediately after sampling and during transport and kept refrigerated until thin-sectioning, which was conducted immediately after being brought to room temperature and hydration. We acknowledge that any change from immediate conditions in the field needs to be minimized, and refrigeration

was therefore chosen because alternatives (air or oven drying, freeze-drying, or freezing) are known to induce changes specifically in the architecture of mineral-organic, organic-organic, and pore space spatial distribution (Kaiser et al., 2015; Kim and Choi, 2018), which was the primary focus of this work. The refrigeration temperature of 9°C is a maximum value listed to give a conservative estimate of the possible temperature experienced by the soil given field sampling, transportation, and fluctuations in the refrigeration unit: the temperature range is closer to 4-9°C, which was added to the revision.

Was the soil sampled undisturbed?

Response: While any manipulation to soil aggregates is likely to distort their dimensions to some degree, the cryogenic sectioning methods used were designed to minimize disturbance to the spatial arrangement and chemical composition at the scale relevant to the analysis of organo-mineral and organo-organic interfaces (i.e., within an intact microaggregate). The potential for disturbance via thin sectioning with aggregates stabilized using rapidly frozen water is a tradeoff for using stabilizing resins containing carbon (that may obscure the composition of native OM). At larger scales, bulk samples were sieved and otherwise disturbed during sampling, which we made sure is described in the revision (lines 375-378). Previous studies by others and in our group (Kinyangi et al., 2006; Wan et al., 2007; Lehmann et al., 2008; Lehmann and Solomon, 2010) showed no spatial re-arrangement at the micro-aggregate scale through the type of sample handling we conducted.

[4] What was the methodology of the quantification.? There seems to be almost zero info about this. What was the variability in the measurements? Where are the confidence intervals? What was the sampling procedure on the thin sections - I note some info but not much and certainly not about the placement of the measuring grids.

Response: Thank you for posing an important point that relates to the nature of high-resolution measurements: as the spatial resolution and information on chemical composition of a technique increases, the sample throughput correspondingly decreases. However, direct visual evidence for phenomena that occur at the very fine scale can only be achieved with techniques that enable very high spatial resolution measurements, providing essential, unambiguous evidence for their existence. As such, high resolution measurements uniquely provide two types of insights in our context: (1) to confirm that a mechanism hypothesized at larger scale is actually present in specific location; and (2) to generate a hypothesis that a process may be important as it is unambiguously observed, and therefore should be studied at larger scale. This essential benefit of high-resolution measurements was highlighted in the revision (lines 86-89). A measurement grid, as would be used for optical microscopy, is not applicable to STEM-EELS measurements at this scale. The sample analysis pipeline is now outlined in Response Figure 3 (and used as a new Supplementary Fig. A7 in the revision), and the method for selection of regions of analysis was incorporated into the methods section of the revision (lines 437-441). With respect to comparisons among multiple samples, statistical analysis is not possible at this point due to the significant reduction in throughput of this technically extremely challenging experimental workflow, with

measurements limited to usually a few samples. Consequently, potential assessment of variability is limited to within each image: analysis of different image regions could potentially provide a “confidence interval” or estimate of error, but displaying within-image error is not based on true replication and as such could be misleading. Further, variability within an image could be a function of true sample heterogeneity rather than random variation.

Response Figure 3. Analysis workflow illustrating selection of regions for detailed, high-resolution analysis of mineral-organic and organo-organic regions of interest.

Summary

This paper is very much focussed on a range of techniques that potentially could provide new insights to soil at the nano scale. However, it fails on a number of important points from the original, unstated, hypothesis through to sampling and onto analysis. I sense that it's from a PhD that focusses on techniques rather than quantification. My recommendation is "Reject".

Response: We revised the introduction and key portions of the discussion to make our points clearer and to be more explicit about our objectives and hypotheses. We set out our study with a clear goal (that we hope to have made clearer in this revision) of finding out what the chemical composition of the organo-mineral interface (literally) looks like.

Our previous research (as stated above) found confirmation of zonal arrangement with distance from the mineral surface (confirming the model by Kleber et al., 2007 Biogeochemistry). However, the spatial resolution of 30nm and 50nm in our earlier work did not give us confidence that we were able to observe the interface. We therefore needed to improve the spatial resolution of our method.

That the goal could only be achieved by a large methodological effort and accomplishment by a graduate student, should not be a detriment of this study, but is in our view an asset, as we are sure the referee agrees.

Through this work presented here, we discovered the unordered rather than ordered structure that included different organic phases. We had not expected these clear phases of distinctly different organic forms that are spatially separated, as we had shown earlier clear gradients (see figure below). Even if one may hypothesize mixtures also at the nano-meter scale, two aspects make the results noteworthy: (1) even if we may hypothesize mixtures and interfaces, these have not yet been shown using a technique that gives information about organic functional group composition at single-digit nm resolution to actually 'observe' the interface; and (2) the distinct spatial arrangement of clear patchiness was not expected. We hope to have made this clearer in the revision.

Figure taken from Lehmann J and Solomon D 2010 Organic carbon chemistry in soils observed by synchrotron-based spectroscopy. In B Singh and M Gräfe (eds) Synchrotron-based Techniques in Soils and Sediment. Elsevier, Amsterdam, pp 289-312.

Response Literature Cited

- Chadwick, O. A., Kelly, E. F., Hotchkiss, S. C. & Vitousek, P. M. Precontact vegetation and soil nutrient status in the shadow of Kohala Volcano, Hawaii. *Geomorphol.* **89**, 70–83 (2007).
- Cotrufo, M. F., Wallenstein, M. D., Boot, C. M., Deneff, K. & Paul, E. The Microbial Efficiency-Matrix Stabilization (MEMS) framework integrates plant litter decomposition with soil organic matter stabilization: do labile plant inputs form stable soil organic matter? *Global Change Biol.* **19**, 988–995 (2013).
- Coward, E. K., Ohno, T. & Plante, A.F. Adsorption and molecular fractionation of dissolved organic matter on iron-bearing mineral matrices of varying crystallinity. *Environ. Sci. Technol.* **52(3)**, 1036–1044 (2018).
- Deng, Y. & Dixon, J. B. Soil organic matter and organic-mineral interactions. In *Soil Mineralogy with Environmental Applications, SSSA Book Series No. 7.* (eds Dixon, J.B. & Shultze, D.G.) (Soil Science Society of America, Madison, WI, 2002).
- Edwards, A. P. & Bremner, J. M. Microaggregates in soils. *J. Soil Sci.* **18**, 64–73 (1967).
- Eswaran, H., et al. Global carbon stocks. In *Global Climate Change and Pedogenic Carbonates* (eds Lal, R., et al.) 13-26 (Lewis Publishers, 2000).
- Grant, K. E., Galy, V. V., Chadwick, O. A. & Derry, L. A. Thermal oxidation of carbon in organic matter rich volcanic soils: insights into SOC age differentiation and mineral stabilization. *Biogeochem.* **144**, 291–304 (2019).
- Hachtel, J. A., et al. Identification of site-specific isotopic labels by vibrational spectroscopy in the electron microscope. *Science* **363(6426)**, 525-528 (2019).
- Heckman, K., Throckmorton, H., Horwath, W. R., Swanston, C. W. & Rasmussen, C. Variation in the molecular structure and radiocarbon abundance of mineral-associated organic matter across a lithosequence of forest soils. *Soil Syst.* **2(36)**, 2020036 (2018).
- Hitchcock, A. P., et al., 2019. Spatially resolved soft X-ray spectroscopy in scanning X-ray microscopes. *Micros. Microanal.* **25(Suppl 2)**, 254–255 (2019).
- Inagaki, T. M., et al. Subsoil organo-mineral associations under contrasting climate conditions. *Geochim. Cosmochim. Acta* **270**, 244–263 (2020).
- Jokisaari, J. R. et al. Vibrational spectroscopy of water with high spatial resolution. *Adv. Mater.* **30(36)**, 1802702 (2018).

- Kaiser, M., Kleber, M. & Berhe, A. A. How air-drying and rewetting modify soil organic matter characteristics: An assessment to improve data interpretation and inference. *Soil Biol. Biochem.* **80**, 324–340 (2015).
- Kallenbalch, C. M., Frey, S. D. & Grandy, A. S. Direct evidence for microbial-derived soil organic matter formation and its ecophysiological controls. *Nature Commun.* **7**, 13630 (2016).
- Kazanski, C. E., Riggs, C. E., Reich, P. B. & Hobbie, S. E. Long-term nitrogen addition does not increase soil carbon storage or cycling across either temperature forest and grassland sites on a sandy outwash plain. *Ecosyst.* **22**, 1592–1605 (2019).
- Keiluweit, M., et al. Nano-scale investigation of the association of microbial nitrogen residues with iron (hydr)oxides in a forest soil O-horizon. *Geochim. Cosmochim. Acta* **95**, 213–226 (2012).
- Kelly, E. F., Chadwick, O. A. & Hilinski, T. E. The effect of plants on mineral weathering. *Biogeochem.* **42**, 21–53 (1998).
- Kim, E-A. & Choi, J. H. Changes in the mineral element compositions of soil colloidal matter caused by a controlled freeze-thaw event. *Geoderma* **318**, 160-166 (2018).
- Kinyangi, J., et al. Nanoscale biogeochemical complexity of the organomineral assemblage in soil: Application of STXM microscopy and C 1s-NEXAFS spectroscopy. *Soil Sci. Soc. Am. J.* **70**, 1708–1718 (2006).
- Kleber, M., Mikutta, R., Torn, M. S. & Jahn, R. Poorly crystalline mineral phases protect organic matter in acid subsoil horizons. *Eur. J. Soil Sci.* **56**, 717-725 (2005).
- Kleber, M. et al. Mineral-organic associations: formation, properties, and relevance in soil environments. *Adv. Agron.* **130**, 1–140 (2015).
- Kleber, M., Sollins, P. & Sutton, R. A conceptual model of organo-mineral interactions in soils: self-assembly of organic molecular fragments into zonal structures on mineral surfaces. *Biogeochem.* **85**, 9–24 (2007).
- Knicker, H. Soil organic N – An under-rated player for C sequestration in soils? *Soil Biol. Biochem.* **43(6)**, 1118–1129 (2011).
- Kramer, M. G., Sanderman, J., Chadwick, O. A., Chorover, J. & Vitousek, P. M. Long-term carbon storage through retention of dissolved aromatic acids by reactive particles in soil. *Glob. Chang. Biol.* **18**, 2594–2605 (2012).

- Lawrence, D., et al. Technical Description of Version 5.0 of the Community Land Model (CLM) (National Center for Atmospheric Research, University Corporation for Atmospheric Research, Boulder, CO, 2018).
- Lehmann, J., et al. Spatial complexity of soil organic matter forms at nanometre scales. *Nature Geosci.* **1**, 238–242 (2008).
- Lehmann, J., Kinyangi, J. & Solomon, D. Organic matter stabilization in soil microaggregates: implications from spatial heterogeneity of organic carbon contents and carbon forms. *Biogeochem.* **85**, 45–57 (2007).
- Lehmann, J. & Solomon, D. Organic carbon chemistry in soils observed by synchrotron-based spectroscopy. In *Developments in Soil Science*, Vol. 34 (eds McBratney, A.B., & Hartemink, A.E.), 289–312 (Elsevier, 2010).
- Milne, A. E., Lehmann, J., Solomon, D. & Lark, R. M. Wavelet analysis of soil variation at nanometre to micrometer-scales: an example of organic carbon content in a micro-aggregate. *Eur. J. Soil Sci.* **62**, 617–628 (2011).
- Nunan, N., et al. Quantification of the in situ distribution of soil bacteria by large-scale imaging of thin sections of undisturbed soil. *FEMS Microbiol. Ecol.* **37(1)**, 67–77 (2001).
- Rasmussen, C., et al. Beyond clay: towards an improved set of variables for predicting soil organic matter content. *Biogeochem.* **137(3)**, 297–306 (2018).
- Raynaud, X. & Nunan, N. Spatial ecology of bacteria at the microscale in soil. *PLoS ONE* **9(1)**, e8217 (2014).
- Riggs, C. E., Hobbie, S. E., Bach, E. M., Hofmockel, K. S. & Kazanski, C. E. Nitrogen addition changes grassland soil organic matter decomposition. *Biogeochem.* **125**, 203–219 (2015).
- Schmidt, M. W., et al. Persistence of soil organic matter as an ecosystem property. *Nature* **478(7367)**, 49–56 (2011).
- Schoeneberger, P. J., Wysocki, D. W., Behnam, E. C. & Broderson, W. D. *Field Book for Describing and Sampling Soils* (Natural Resources Conservation Service, USDA, National Soil Survey Center, 1998).
- Six, J., Elliott, E. T. & Paustian, K. Soil macroaggregate turnover and microaggregate formation: a mechanism for C sequestration under no-till agriculture. *Soil Biol. Biochem.* **32**, 2099–2103 (2000).
- Sollins, P., et al. Organic C and N stabilization in a forest soil: Evidence from sequential

- density fractionation. *Soil Biol. Biochem.* **38(11)**, 3313–3324 (2006).
- Solomon, D., et al. Micro- and nano-environments of carbon sequestration: Multi-element STXM-NEXAFS spectromicroscopy assessment of microbial carbon and mineral associations. *Chem. Geol.* **329**, 53–73 (2012).
- Steffens, M., et al. Identification of distinct functional microstructural domains controlling C storage in soil. *Environ. Sci. Technol.* **51(21)**, 12181–12189 (2017).
- Tisdall, J. M. & Oades, J. M. Organic matter and water-stable aggregates in soils. *J. Soil Sci.* **33**, 141–163 (1982).
- Torn, M. S., Trumbore, S. E., Chadwick, O. A., Vitousek, P. M. & Hendricks, D. M. Mineral control of soil organic carbon storage and turnover. *Nature* **389**, 170–173 (1997).
- Torn, M. S., Swanston, C. W., Castanha, C. & Trumbore, S. E. Storage and turnover of organic matter in soil. In *Biophysico-Chemical Processes Involving Natural Nonliving Organic Matter in Environmental Systems* (eds Senesi, N., Xing, B., Huang, P.M.), 219–272 (John Wiley & Sons, Inc., 2009).
- Wan, J., Tyliszczak, T. & Tokunaga, T. K. Organic carbon distribution, speciation, and elemental correlations within soil microaggregates: applications of STXM and NEXAFS spectroscopy. *Geochim. Cosmochim. Acta*, **71(22)**, 5439–5449 (2007).
- Young, I. M. & Crawford, J. W. Interactions and self-organization in the soil-microbe complex. *Science*, **304(5677)**, 1634–1637 (2004).

REVIEWER COMMENTS

Reviewer #1 (Remarks to the Author):

The authors substantially discussed my comments and improved the manuscript. I welcome the novel figure 1, as it helps readers to follow the argumentation more easily.

Yet also after re-reading I still think that some aspects of the story-line are not fully developed. In my opinion, the following parts of the argumentation need to be clarified:

First, in the reply letter the authors point out that the nm-scale investigated in the present study is smaller than the micrometre-scale assessments of previous publications, but larger than the scale relevant for individual molecules. The second point is important, and I feel it should be discussed in more detail in the manuscript text. In particular, I wondered if presence of very large macromolecules are an alternative explanation of the data (i.e., alternative to occurrence of organo-organo-interfaces). For instance, could the size of proteins or large plant biomolecules such as tannins be in the range of a few nanometres? Also it may be possible that fragments of dead microbial cells or plant litter fall into that range?

The concept that the organic matter accumulation close to mineral surfaces is due to formation of large `structures` by interactions between small molecules, seems to be similar to old views of formation of macromolecular humic substances in soil. Maybe this can be added to the discussion.

In relation to this point, I wonder if the formation of large organic matter structures via organo-organo-interactions may also occur without the presence of reactive mineral-surfaces such as AlOH surfaces. Because if organo-organo-interactions can occur a few nanometres in distance to the mineral, what is the mechanistic role of the mineral surface (or in other words: how does the mineral surface chemistry affect reactions that are so far away from the mineral surface?). I think the data may imply that organo-organo-interfaces also form without the reactive minerals.

The authors assume that „role of polyphenol complexation of N-rich compounds“ (line 258) may be an explanation of occurrence of organo-organo-interfaces. Accordingly, the conclusions in the abstract should include that the input of plant molecules is important (currently only „N-rich residues, including microbial metabolites“ is mentioned, but this seems to be only one key part of the mechanistic framework outlined in the manuscript).

I don't understand the following sentence in figure 4: „Layered structure of distinct functional groups detected at single nanometer rather than micrometer scales“. If earlier studies describe layered structures on a larger scale, it means that layered structures can occur at both, the nm-scale and the larger micrometre scale (i.e., one cannot disprove micrometre scale phenomena by studying the nm-scale). Thus, it needs to be explained why such layered structures can occur on different scales. I think this is an interesting and puzzling finding.

Reviewer #2 (Remarks to the Author):

The authors have satisfactorily responded to all reviewer concerns and the finished manuscript is acceptable for publication.

Reviewer #3 (Remarks to the Author):

I thank the authors for their detailed response to the comments. This is greatly appreciated. However,

I remain unconvinced by the statements of the authors that their work reveals any new insights. I note the concept of layering has disappeared (bar one comment on page 3), but really no rationale for that edit. Nor do I really see much in the way of new thinking. As I mentioned before heterogeneity is an inherent property of all soils at some range of scales. They state that at the high resolution they are looking they see, for the first time such variability. But even in the papers they reference for this statement there is clear evidence that such variability has already been observed and quantified:

Kinyangi et al (2005). "Spatial distribution of Carbon forms.. NEXAFS identified nano variability in mineral and C chemical composition of the mineral pore assemblage" It also could be argued that they have linked form and function in a more appropriate way than the current manuscript.

In Solomon et al (2010) with some of the co-authors of this paper, it states. " NEXAFS..clearly demonstrated the existence of spatially distinctzones.'

All done at very fine spatial scales, as in this manuscript.

Indeed, in the current manuscript the lack of appropriate quantification is real: "However, we highlight that the arrangement in space may not only extend linearly away from a mineral surface, but also in multi-dimensional space at the boundaries of irregularly shaped organic patches." Again no layering and this statement seems to say that everything possible? I am not sure what, in a 2-D analysis of soil they can state about multi-dimensionality which I guess is related to 3d and time? They do recognise this limitation in a slight way at the end of the article with a comment on tomography.

I also have a worry about the sampling procedure. Why did the authors not take standard thin sections, whilst conserving the organic-mineral locations (already published) then make ultra thin sections and then look at their very small scales in situ? What they have done is take small bit of soil, not spatially referenced, and then made ultra thin sections. This loses what they are trying to find, spatial relevance across scales.

Just going to a small scale does not make this publishable in Nature Comms. It needs a more resilient hypothesis, linked to function backed up by a theoretical construct. There is almost no function here. It's all form and almost no quantification. Saying that, very well done and worthy of publication in some discipline journal. But this not the next big thing and indeed some has been done before, as mentioned to the authors.

Just going to a small scale does not make this publishable in Nature Comms. It needs a more resilient hypothesis, linked to function backed up by a theoretical construct.. There is almost no function here. It's all form and almost no quantification. Saying that, very well done and worthy of publication in some discipline journal. But this not the next big thing and indeed some has been done before, as mentioned to the authors.

So, it's still a reject from me. I emphasise however, this is very good work, well carried out and, now, very well written. It will find a home in a good quality discipline journal.

Nature Communications: “Organo-organic and organo-mineral interfaces in soil at the nanometer scale”

Revision 2

Response to Review

REVIEWER COMMENTS

Author responses: Thank you for the additional suggestions. Please note that line numbers in the responses below refer to the submitted version without tracked changes.

Reviewer #1 (Remarks to the Author):

The authors substantially discussed my comments and improved the manuscript. I welcome the novel figure 1, as it helps readers to follow the argumentation more easily.

Yet also after re-reading I still think that some aspects of the story-line are not fully developed. In my opinion, the following parts of the argumentation need to be clarified:

First, in the reply letter the authors point out that the nm-scale investigated in the present study is smaller than the micrometre-scale assessments of previous publications, but larger than the scale relevant for individual molecules. The second point is important, and I feel it should be discussed in more detail in the manuscript text.

Response: Thank you for highlighting that the comment about implications of individual molecular interactions was not fully explored in the manuscript. In addition to the point made in the first revision (that the scale of our observations approached a meaningful scale to consider the implications for the “zonal” model), we also now note in the second revision that the scale is still larger than individual functional groups or bonds in lines 201-203 and 250-253. However, in some cases the size and and scale of our measurements could be on the order of some molecules, which we discuss further in responses below.

In particular, I wondered if presence of very large macromolecules are an alternative explanation of the data (i.e., alternative to occurrence of organo-organic interfaces). For instance, could the size of proteins or large plant biomolecules such as tannins be in the range of a few nanometres? Also it may be possible that fragments of dead microbial cells or plant litter fall into that range?

Response: Thank you for pointing out the potential range in size for different organic constituents that may be present. At the interface between organic phases (Figs 4 and 5 in the text), the at least <11 and up to <5 nm thick layers observed at the interface would be smaller than large biomolecules such as tannins (20+ nm), but on the order of small proteins (~5 nm). Fragments of microbial or plant cells/organelles (100s-1000 nm) are much larger, and even more so intact particulate (i.e., not “dissolved”) litter (at least >450 nm by definition, but more often >5000 nm) (Response Figure 1). In general, we

consider “organo-organic interactions” to be inclusive of interactions between any form of OM, including biomolecules, and agree with the reviewer that the apparent thickness of layers could be consistent with small biomolecules or associations of biomolecules. We clarified this in this revision, stating: “The observed layers of distinct OM composition are consistent in size with small biomolecules or associations^{23,38} of small biomolecules, such as small proteins. However, even the single-digit nanometer resolution does not allow for the sub-nanometer spatial mapping of individual functional groups or bonds.” (lines 199-203).

Response Figure 1. Relative size of cellular components relative to <10 nm thick layers at organo-organic interface (characterized in Figs 4 and 5 in the text, and conceptually in Fig. 1 in the text).

The concept that the organic matter accumulation close to mineral surfaces is due to formation of large `structures` by interactions between small molecules, seems to be similar to old views of formation of macromolecular humic substances in soil. Maybe this can be added to the discussion.

Response: This question raises an interesting point that relates to the above discussion of biomolecule size and associations of biomolecules, a more generalized concept of “macromolecule” formation pathways; e.g., as discussed in Sutton and Sposito (2005) and Schmidt and Martinez (2018). To address this point, we now note in the text (as noted above) that associations of relatively small (<5 nm) biomolecules may be involved in the formation of layers, and cite these key references (lines 199-201).

In relation to this point, I wonder if the formation of large organic matter structures via organo-organic interactions may also occur without the presence of reactive mineral-surfaces such as AlOH surfaces. Because if organo-organic interactions can occur a few nanometres in distance to the mineral, what is the mechanistic role of the mineral surface (or in other words: how does the mineral surface chemistry affect reactions that

are so far away from the mineral surface?). I think the data may imply that organo-organ interfaces also form without the reactive minerals.

Response: Thank you for highlighting this point, which we agree is an important implication of this data. In the first revision, we noted: “The described organo-organic interfaces point towards the need to consider soil organic C stabilization mechanisms that are independent of soil minerals or mineral types and involve N-rich surfaces”, which alludes to the suggestion that organo-organic interactions may be independent of mineral surface area, chemistry, and other factors controlling direct single-molecule organic C adsorption on mineral surfaces. We have made this point more explicit in this revision by adding: “The described organo-organic interactions may occur at a distance from or separately from mineral surfaces, pointing towards the need to consider soil organic C stabilization mechanisms that are independent of soil minerals (de-emphasizing variables such as mineral surface area or surface chemistry), and involve N-rich surfaces¹³”, in lines 265-269.

The authors assume that „role of polyphenol complexation of N-rich compounds“ (line 258) may be an explanation of occurrence of organo-organ interfaces. Accordingly, the conclusions in the abstract should include that the input of plant molecules is important (currently only „N-rich residues, including microbial metabolites“ is mentioned, but this seems to be only one key part of the mechanistic framework outlined in the manuscript).

Response: Thank you for raising this point, which we agree was not clearly articulated. In this revision, the statement referring specifically to microbial metabolites was removed from the abstract. Because our intention was not to convey a specific source for molecules involved in interactions, the two references in the text that specifically (“plant-derived cytokinins” in line 222) or indirectly (polyphenols in line 270) mention plant-derived materials were revised for clarity. Specifically, “plant-derived cytokinins” (as an example of a compound with relatively high aliphatic C, total C and O, and low N) was revised to simply “cytokinins”, because these compounds can be produced by both plants and microbes (Akhtar et al., 2019). The reference to polyphenol complexation of N-rich compounds was intended to provide an example of “organo-organic interactions” (as we refer to in the manuscript), but not to invoke a specific interaction mechanism (which is not directly addressed by our data); we have revised this point to state, “In litter decomposition, organo-organic interactions between aromatic and N-rich compounds have been linked to slower litter turnover⁴⁰. Here, we suggest interactions between relatively low and high-N OM forms may also be relevant for soil OM” (lines 169-272).

I don't understand the following sentence in figure 4: „Layered structure of distinct functional groups detected at single nanometer rather than micrometer scales“. If earlier studies describe layered structures on a larger scale, it means that layered structures can occur at both, the nm-scale and the larger micrometre scale (i.e., one cannot disprove micrometre scale phenomena by studying the nm-scale). Thus, it needs to be

explained why such layered structures can occur on different scales. I think this is an interesting and puzzling finding.

Response: Thank you for raising this important point. We agree with the referee that the existence of layered structures of nanometers in size per se does not exclude the existence of layers micrometers in size (i.e., it is indeed theoretically possible that layered and organized gradients exist at both scales). In this study, irregularity and patchiness of C and N forms approximately 1 micrometer or less in size were clearly resolved and statistically supported (using MCR): this direct observation proves their existence in this sample. We admit, we cannot state unequivocally that the patchiness observed here did or did not exist in previous studies (including our own), because generally speaking, the lower spatial resolution and typically much larger field of view likely obscured the existence of such patchiness, if it was present. However, such evidence of previous studies at lower resolution implied ordered gradients,- and this conclusion informed our traditional understanding of the properties of organo-mineral interfaces. To be fair, these previous studies frequently acknowledge that their spatial resolution was too coarse to unequivocally conclude what the interface looks like,- yet they were not able to point at the evidence we are able to show.

Consequently, the irregularity and patchiness of C and N forms that we observed here provides both a contrast to previous observations and an insight into a proposed new (and we argue, more accurate) representation of the spatial arrangement of SOM. Specifically, we show that the N content and certain C and N forms change vastly up and down within what was assumed to be a uni-directional and ordered gradient. We have revised the Figure 1 caption to clarify that we directly identified heterogeneous patches, which contradicts previous assumptions of organized and uni-directional layering with distance to the mineral. In particular, the Figure 1 caption was amended to read: "These observations directly reveal heterogeneous patches that contradict previous assumptions of ordered and uni-directional layering of OM forms, generating new insights into OM composition at mineral-organic and organo-organic interfaces."

Reviewer #2 (Remarks to the Author):

The authors have satisfactorily responded to all reviewer concerns and the finished manuscript is acceptable for publication.

Response: Thank you for your feedback and support of the manuscript.

Reviewer #3 (Remarks to the Author):

I thank the authors of their detailed response to the comments. This is greatly appreciated. However, I remain unconvinced by the statements of the authors that their work reveals any new insights. I note the concept of layering has disappeared (bar one comment on page 3), but really no rationale for that edit. Nor do I really see much in the way of new thinking.

Response: Thank you for your feedback and consideration of the manuscript. In the first revision, we aimed to refine the discussion of layering with respect to the boundaries between organic rather than a more simplistic contrast of “layered” vs. “patchy” accumulation. The background for this concept was developed in lines 52-63 of the first revision (R1 version without tracked changes), and mentioned in particular as a critical finding in lines 176-182, in the discussion lines 240-244, and in Figs 1 and 5. To reinforce the critical considerations of scale and size of “layers”, in this second revision, we added text to the abstract as well, stating: “Using cryo-electron microscopy with electron energy loss spectroscopy (EELS), we show organo-organic interfaces in contrast to exclusively organo-mineral interfaces. Single-digit nanometer-size layers of C forms were detected at the organo-organic interface, showing alkyl C and nitrogen (N) enrichment (by 4 and 7%, respectively).” This also highlights one aspect of the novelty of this work, in our view: we directly visualize and describe the nature of interfaces not only between organic matter and mineral surfaces, but also between organic constituents of different composition.

As I mentioned before heterogeneity is an inherent property of all soils at some range of scales. They state that at the high resolution they are looking they see, for the first time such variability. But even in the papers they reference for this statement there is clear evidence that such variability has already been observed and quantified:

Kinyangi et al (2005). "Spatial distribution of Carbon forms.. NEXAFS identified nano variability in mineral and C chemical composition of the mineral pore assemblage" It also could be argued that they have linked form and function in a more appropriate way than the current manuscript.

In Solomon et al (2010) with some of the co-authors of this paper, it states. " NEXAFS..clearly demonstrated the existence of spatially distinctzones."

All done at very fine spatial scales, as in this manuscript.

Response: Thank you for highlighting the variable scales and concepts discussed in the cited manuscripts. One source of potential ambiguity is the way in which the term “nano” is used. In Kinyangi et al. (2006) and Solomon et al. (2012), “nano” variability refers to sub-micrometer variability, on the order of 100s of nanometers; for example, in Kinyangi et al. (2006), “nanoscale pore constrictions” are > 500 nm in size. Similarly, Solomon et al. (2012) refers to “nano-C repository zones” described at a sub-micrometer scale, but still several orders of magnitude larger than the scale considered here in this new manuscript. Both references use a spot size of 40-50 nm (the lower limit of spatial resolution), ~20 times larger than the highest resolution used here (2 nm, in Figure 5). The increase in resolution is not just a methodological advancement or a better image of the same structure, but allows to distinguish new structures (the interface between two organic forms) that were previously impossible to definitively resolve. We agree that the terminology of the “mineral pore assemblage” in Kinyangi et al. (2006) may be ambiguous with respect to the scale considered, so in the text we have made it more explicit that observations at the scale of aggregate cross-sections and pores have

shown patchy distribution, while layered architecture and organized gradients have been shown with SOM coatings of mineral assemblages as a function of distance to the mineral surface (lines 54-56). These layers as a function of distance to the mineral surface were, as mentioned above, implied by the evidence obtained from microscopy with a spatial resolution of 40-50 nm. Additionally, we added “pore structures” to Figure 1a to emphasize previous observations of heterogenous distribution.

Indeed, in the current manuscript the lack of appropriate quantification is real: "However, we highlight that the arrangement in space may not only extend linearly away from a mineral surface, but also in multi-dimensional space at the boundaries of irregularly shaped organic patches." Again no layering and this statement seems to say that everything possible? I am not sure what, in a 2-D analysis of soil they can state about multi-dimensionality which i guess is related to 3d and time? They do recognise this limitation in a slight way at the end of the article with a comment on tomography.

Response: Thank you for highlighting potential ambiguity with the term “multidimensional”: we agree that this may be read as 3-D descriptions or another dimension such as time, which was not the intent. Rather, we aimed to emphasize the distinction between a uni-directional architecture and irregularly shaped organic patches, and agree that more explicit statement of layering here benefits the discussion mentioned above. As such, we have removed the term “multidimensional” and added the text: “... arrangement in space may not only extend linearly in organized layers away from a mineral surface, but also at the boundaries of irregularly shaped organic patches (Fig. 1d, e, f)” (lines 254-256).

I also have a worry about the sampling procedure. Why did the authors not take standard thin sections, whilst conserving the organic-mineral locations (already published) then make ultra thin sections and then look at their very small scales in situ? What they have done is take small bit of soil, not spatially referenced, and then made ultra thin sections. This loses what they are trying to find, spatial relevance across scales.

Response: Thank you for raising this important methodological consideration. In our sample preparation approach, we prepared initial aggregate thin sections (1-5 micrometer in thickness) prior to FIB thin sectioning using frozen water as a stabilizing medium (described in Methods 3.2). In this revision, we expanded the first sentence of this section to better clarify the sequence of sample preparation, stating: “Cryogenic thin-sectioning (to approximately 1-5 μm thickness) was undertaken to improve efficiency and selection of regions for subsequent cryo-focused ion beam (FIB) milling to electron transparency”. The use of water rather than standard carbon-based stabilizing materials for thin-sectioning allows us to detect composition differences in native soil organic carbon, a critical technical advancement and benefit of this approach. We have made this consideration more explicit in the revision in Methods Section 3.1, now stating: “To be able to resolve native soil OM composition, thin-sectioning without use of stabilizing materials that interfere with OM detection provides a technical challenge and area for methodological advancement”. Consequently, our approach

does indeed preserve the spatial location of areas that were analyzed within the soil aggregate. We do agree that elemental mapping at a coarse scale prior to further thin-sectioning would be advantageous; in this work, the benefit of cryogenic approaches (limiting damage and preserving soil structure without use of interfering resins) comes with the requirement that samples must stay frozen continuously, which limits the use of complementary coarse-scale imaging or other techniques. However, in future work, a hierarchical imaging and spectroscopy approach with increasingly higher resolution measurements could be further explored. We now state the need for this in future work, now noted in lines 312-314.

Just going to a small scale does not make this publishable in Nature Comms. It needs a more resilient hypothesis, linked to function backed up by a theoretical construct. There is almost no function here. It's all form and almost no quantification. Saying that, very well done and worthy of publication in some discipline journal. But this is not the next big thing and indeed some has been done before, as mentioned to the authors.

So, it's still a reject from me. I emphasize however, this is very good work, well carried out and, now, very well written. It will find a home in a good quality discipline journal.

Response: Thank you for the supporting comments regarding the quality of the implementation of the work and the manuscript. As we note in the text, "Single-digit nanometer-scale imaging and spectroscopy techniques may enable us to confirm the existence of previously hypothesized nanoscale interactions, and generate novel hypotheses to be tested at larger scales based on observations of previously inaccessible structures." There are ways in which this work is consistent with existing theoretical constructs (but benefits them in providing a new level of evidence at very high spatial resolution that were hitherto only hypothesized, but could not be directly proven). In the case of spatial architecture of OM, our observations in important ways support the theoretical framework of the Kleber (2007) "zonal" model. The observation of at least <10 nm thick layers at the organo-organic interface do not contradict the existence of discrete "zones" (i.e., layers of OM composition), but significantly expand them. However, we also highlight differences in composition specific to the demonstrated organo-organic interfaces in contrast to only organo-mineral interfaces, which to our knowledge is a new direction for conceptual expansion.

In addition, we discuss the framework that preferential reactions with oxidized functional groups and N-containing biomolecules are involved in the formation of associations between reactive Fe and Al and SOM. In this work, we provide unambiguous evidence for the co-location of N-rich OM with Al mineral surfaces (since no imaging at this spatial scale was done for soil organic matter before, this is the first direct evidence). However, we also generated new directions for future exploration, including: (1) further evaluation of interaction mechanisms specific to N-substituted carboxylic groups; (2) potential for either competitive or synergistic interactions with co-occurrence of inorganic N accumulation and N-rich organic matter at the interface; and (3) reactive Al-specific interaction with N-rich OM. These provide in our view motivation for new inquiry.

Response References:

Akhtar, S. S., Mekureyah, M. F., Pandey, C., & Roitsch, T. Role of cytokinins for interactions of plants with microbial pathogens and pest insects. *Front Plant Sci.* **10**, 1777 (2019).

Schmidt, M. P. & Martinez, C. E. Supramolecular association impacts biomolecule adsorption onto goethite. *Environ. Sci. Technol.* **52**, 4079–4089 (2018).

Sutton, R., & Sposito, G. Molecular structure in soil humic substances: the new view. *Environ. Sci. Technol.* **39**, 9009-9015 (2005).

REVIEWERS' COMMENTS

Reviewer #3 (Remarks to the Author):

A well written reply.

Response to Reviewers

REVIEWERS' COMMENTS

Reviewer #3 (Remarks to the Author):

A well written reply.

Author response: Thank you for the support of the manuscript. We are grateful for the conversations that have greatly advanced and informed this work.